# Dynamic m⁶A methylation facilitates mRNA triaging to stress granules

Maximilian Anders[1],*, Irina Chelysheva[1],*, Ingrid Goebel[1], Timo Trenkner[1], Jun Zhou[2], Yuanhui Mao[2], Silvia Verzini[1], Shu-Bing Qian[2], Zoya Ignatova[1]

**Reversible post-transcriptional modifications on messenger RNA emerge as prevalent phenomena in RNA metabolism. The most abundant among them is N⁶-methyladenosine (m⁶A) which is pivotal for RNA metabolism and function; its role in stress response remains elusive. We have discovered that in response to oxidative stress, transcripts are additionally m⁶A modified in their 5′ vicinity. Distinct from that of the translationally active mRNAs, this methylation pattern provides a selective mechanism for triaging mRNAs from the translatable pool to stress-induced stress granules. These stress-induced newly methylated sites are selectively recognized by the YTH domain family 3 (YTHDF3) "reader" protein, thereby revealing a new role for YTHDF3 in shaping the selectivity of stress response. Our findings describe a previously unappreciated function for RNA m⁶A modification in oxidative-stress response and expand the breadth of physiological roles of m⁶A.**

## Introduction

Adequate reprogramming of metabolic activities by environmental stress or suboptimal growth conditions is crucial for cell survival. In eukaryotes, the most potent stress-induced inhibition of translation is repression of translational initiation by kinases-induced phosphorylation of Ser51 of eIF2α, which selectively represses translation of mRNAs with cap-dependent translation (Sonenberg & Hinnebusch, 2009). Translationally stalled mRNAs are deposited in stress granules (SGs); membraneless RNA-protein particles (RNPs) that along with non-translating mRNAs associated with translation initiation complexes contain RNA-binding proteins and proteins with low complexity or intrinsically disordered regions (IDRs) (Buchan & Parker, 2009; Kedersha et al, 2013; Jain et al, 2016; Protter & Parker, 2016). SGs are dynamic structures whose assembly is modulated by protein–protein interactions frequently involving their IDRs, whereas translationally stalled mRNAs serve as scaffolds for attachment of the RNA-binding proteins (Panas et al, 2016; Protter & Parker, 2016). SGs are heterogeneous in structure, with denser parts (termed cores) linked by less concentrated regions (termed shells) (Protter & Parker, 2016). In mammalian cells, quantification of the RNA constituents of the SG cores by RNA-sequencing (RNA-Seq) revealed that the targeting efficiency for different mRNAs largely varies and a relatively small fraction of bulk mRNAs accumulates in them (Khong et al, 2017). Although through such systemic approaches our knowledge of protein and mRNA components of the SG cores is steadily increasing (Jain et al, 2016; Khong et al, 2017), whether the composition of the shell differs from that of the core remains elusive.

A long-standing question is the mechanism by which cellular mRNAs are selected in SGs. A prevailing hypothesis has been that stress-induced stalling of cap-dependent initiation is a major sorting factor for triaging mRNAs into SGs, which is consistent with their composition (Buchan & Parker, 2009; Sonenberg & Hinnebusch, 2009; Kedersha et al, 2013). SGs typically contain poly(A) mRNAs, 40S ribosomal subunits, and different initiation factors, and are sensitive to drugs that impair translation initiation, although the composition can vary depending on the type of stress (Kedersha et al, 2005; Buchan & Parker, 2009; Protter & Parker, 2016). For example, initiation factors necessary for assembly of the preinitiation complex might be absent in SGs, arguing that not all mRNAs within the SGs are stalled at initiation (Kedersha et al, 2005). Furthermore, various types of stresses inhibit translation downstream of initiation (Liu et al, 2013). A recent study reveals that mRNA clients of the mammalian SG cores are enriched in species with longer coding and UTR regions and suggests that inefficient translation and/or poor association with ribosomes facilitates association with RNA-binding proteins and consequently with SGs (Khong et al, 2017). However, the full population of mRNAs in SGs, including those from the more mobile periphery, and more specifically mechanistic details underlying mRNA selection and triaging to SGs have not been addressed.

Under various types of stress, a sizeable subset of genes escapes global kinase(s)-dependent inhibition of translation and remains

[1]Institute for Biochemistry and Molecular Biology, Department of Chemistry, University of Hamburg, Hamburg, Germany  [2]Division of Nutritional Science, Cornell University, Ithaca, NY, USA

Correspondence: zoya.ignatova@uni-hamburg.de
Jun Zhou's present address is School of Life Science and Technology, China Pharmaceutical University, Nanjiing, China.
*These authors contributed equally to this work.

translationally active and likely shapes the stress-dependent gene expression at the level of translation. A recently discovered mechanism of mRNA modification in 5′ UTRs offers an attractive solution for this central conundrum: a prevalent methylation at position 6 of adenosine (m⁶A) in the 5′ UTRs enables cap-independent translation (Meyer et al, 2015; Zhou et al, 2015; Coots et al, 2017). m⁶A is the most abundant post-transcriptional modification on mRNA (Roundtree et al, 2017) and is crucial for RNA metabolism, including RNA stability (Wang et al, 2014), splicing (Dominissini et al, 2012; Liu et al, 2015), miRNA processing (Alarcon et al, 2015a, b), mediating translation under normal growth and heat stress (Meyer et al, 2015; Wang et al, 2015; Zhou et al, 2015), or facilitating repair of ultraviolet-induced DNA damage sites (Xiang et al, 2017). m⁶A in mRNAs is reversibly installed at conserved DRACH motif (D = A/G/U; R = A/G; and H = U/A/C) (Meyer & Jaffrey, 2017) by the "writer" complex (methyltransferase like 3 [METTL3], methyltransferase like 14 [METTL14], and Wilms' tumor 1-associating protein [WTAP]) and reversed by demethylases termed "erasers" (fat mass and obesity-associated protein [FTO] and AlkB homologue 5 [ALKBH5]) (Roundtree et al, 2017). Under permissive growth, 3′ UTRs exhibit the highest m⁶A modification levels, which mainly control mRNA stability (Meyer et al, 2012; Wang et al, 2014). The coding sequences (CDSs) are the largest segments in mRNA and statistically contain the most DRACH motifs; however, under physiological conditions, they are poorly m⁶A modified as compared with the 3′ UTRs. This raises two fundamental questions: Are m⁶A modifications in the CDSs dynamic and maybe stress related? Do they contribute to the dynamics of SGs and play a role in the selection of mRNAs in SGs?

Herein, combining systemic deep-sequencing–based approaches with single transcript analysis, we investigated the dynamics of m⁶A modification under mild (200 μM arsenite [AS]) and severe (500 μM AS) oxidative stresses in mammalian cells. We use a cross-linking approach that stabilizes SGs and allows for isolation of the whole SG particles including their mobile shells. Along with mRNAs with stalled initiation complexes, which signal the SG association, in a large set of SG mRNA clients we detected a pervasive m⁶A modification. Our results suggest a role of m⁶A in selectively triaging mRNAs to SGs.

## Results

### Oxidative stress induces additional methylation in mRNAs

To dissect the dynamics of m⁶A modification under oxidative stress, we took advantage of two different cell lines that stably express SG marker proteins: U2OS-G3BP1 (Ohn et al, 2008) and HEK-TIA1 (Damgaard & Lykke-Andersen, 2011). These SG marker proteins, G3BP1 and TIA1, are GFP- or FLAG-tagged, respectively, which enables immunofluorescent detection of SGs. To elicit oxidative stress, we used AS. In both cell lines, SGs formed in a dose-dependent manner (Fig S1A) and contained both mRNAs and proteins (Fig S1B and Table S1) previously identified in SGs (Jain et al, 2016; Khong et al, 2017). The maximal stress dose (500 μM AS) and exposure (30 min) we used caused only marginal changes in the total mRNA levels as revealed by RNA-Seq (Fig S1C). Overall,

comparing with the total mRNAs detected under permissive growth (reads per kilobase per million reads more than the spike-in threshold), we detected only a 6.5% decrease in the total mRNAs under stress (Fig S1C), consistent with previous observations that short AS exposure does not trigger a global transcriptional response and alters the stability of few specific mRNAs (Andreev et al, 2015). Gene ontology (GO) enrichment analysis of the genes degraded under stress showed enriched categories (enrichment score: 7.67) related to transcription (fold enrichment: 1.94; $P$ = 7.88 × 10⁻⁸), including "regulation of transcription" (fold enrichment: 2.01; $P$ = 8.28 × 10⁻⁷), "transcription factor activity" (fold enrichment: 2.11; $P$ = 3.12 × 10⁻⁵), and "DNA binding" (fold enrichment: 1.64; $P$ = 5.56 × 10⁻⁴). Two mRNAs were significantly up-regulated under oxidative stress: immediate early response protein 2 (*IER2*) and *FOS* transcription factor, which are usually up-regulated by environmental cues that increase intracellular levels of reactive oxygen species (Cekaite et al, 2007).

Using anti-m⁶A antibodies, we found pervasive colocalization of the m⁶A-modified RNAs with SGs irrespective of the type of stress, for example, heat or oxidative stress (Fig 1A). Impairment of the "writer" complex by the combined knockdown of METTL3, METTL14, and WTAP proteins markedly decreased the colocalization of the m⁶A signal within SGs, although overall the SGs were visible through the SG marker protein G3BP1 (Fig 1B). It should be noted that a weak m⁶A signal was still detectable because the "writer" complex was silenced to 70% (Fig S2A). METTL3 is the catalytically active subunit of the complex (Roundtree et al, 2017), and its depletion alone led to a decrease in the colocalization m⁶A signal within the SGs (Fig 1B). Generally, the "writers" and "erasers" resided in the nucleus under permissive growth (Fig S2B and C), where the m⁶A modification is believed to primarily take place (Roundtree et al, 2017). In response to oxidative stress, the localization pattern of METTL14, WTAP, and the "erasers" remained unaltered (Fig S2B and C). In contrast, METTL3 partitioned between the nucleus and the cytosol (Fig S2B), resembling recent observations in human cancer cells and mouse embryonic stem cells (Alarcon et al, 2015b; Lin et al, 2016). The functional cooperativity between METTL3 and METTL14 (Wang et al, 2016) requires both readers for methylation of the DRACH motifs. Thus, we cannot firmly conclude that stress-induced changes in m⁶A modifications on mRNA take place directly in the cytosol, although they may because traces of METTL14 are also detectable in the cytosol (Fig S2B).

Because m⁶A modulates mRNA stability (Wang et al, 2014; Mauer et al, 2017), we next determined the global effect of the silenced "writer" complex on the mRNA abundance using RNA-Seq. Overall, comparing with the total mRNAs detected under permissive growth, we did not detect substantial changes in the global mRNA levels following knockdown of the "writer" complex (Fig 1C).

Surprisingly, when analyzing the total RNA with m⁶A antibody, we noticed that the level of the m⁶A signal increased following stress exposure and exhibited a slight stress-dose dependence in both cells (Figs 1D and S2D), suggesting stress-induced increase of m⁶A modifications. The first nucleotide in the m⁷G cap is 2′-*O*-methyladenosine (Aₘ), which can be further methylated at the N⁶ position (m⁶Aₘ) and along with the m⁶A is targeted by the m⁶A antibodies (Mauer et al, 2017). Furthermore, a large fraction of non-coding RNAs (e.g., rRNAs) can also be m⁶A modified (Pan, 2013) and recognized by the m⁶A antibodies. Thus, a large proportion of the m⁶A signal from the total RNA

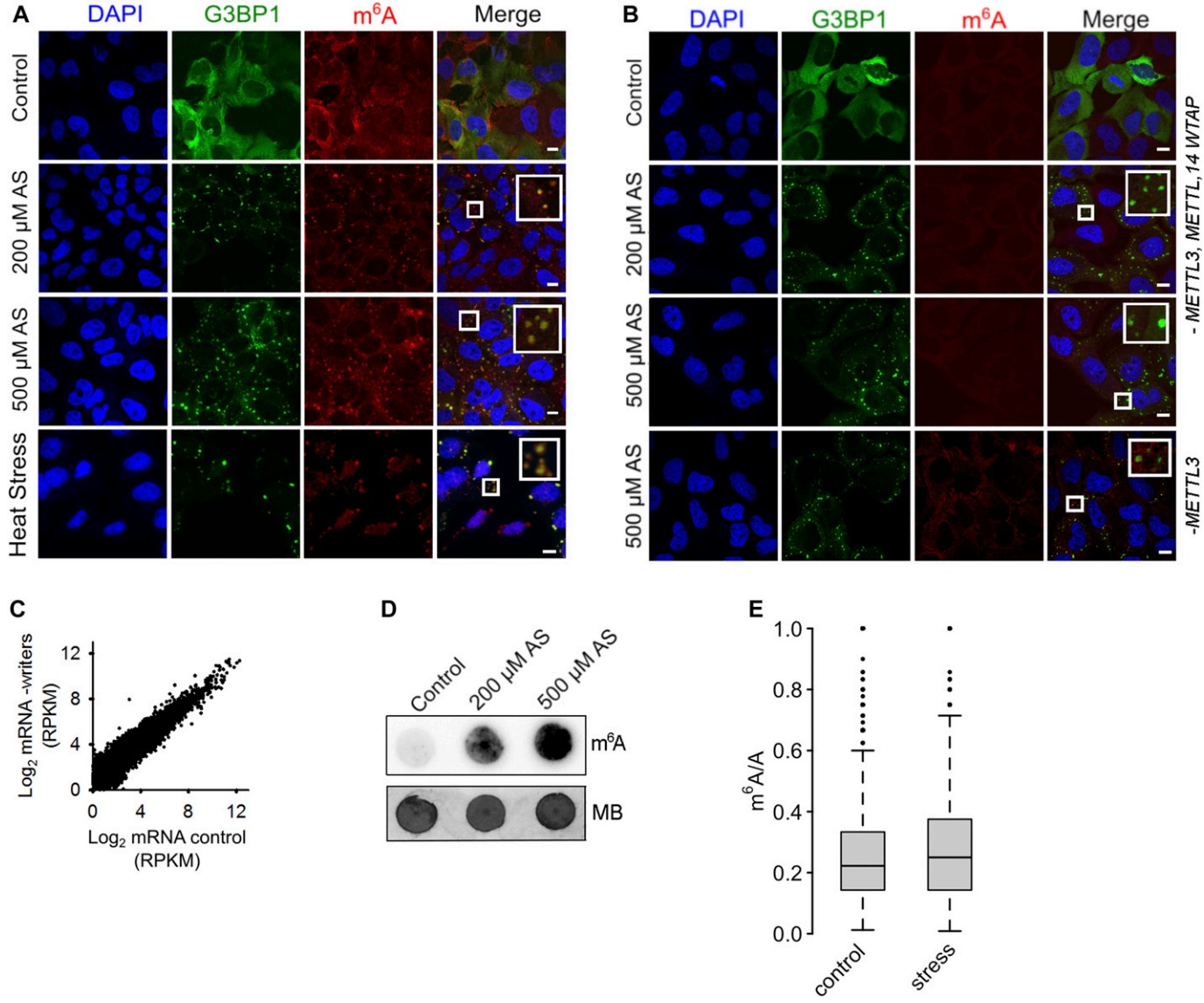

**Figure 1. m⁶A signal increases in response to oxidative stress and accumulates in SGs.**
**(A)** U2OS-G3BP1 cells grown under permissive conditions (control), exposed to mild (200 $\mu$M AS) or harsh (500 $\mu$M AS) oxidative stress for 30 min, or to heat for 2 h at 42°C and immunostained with anti-m⁶A antibody. SGs (hyperfluorescent loci) were visualized through G3BP1–GFP (green), nuclei were counterstained with DAPI (blue). Scale bar, 10 $\mu$m. **(B)** Combined siRNA knockdown of the "writer" complex (*METTL3, METTL14,* and *WTAP*) or *METTL3* alone (lowest row) in U2OS-G3BP1 cells. *METTL3, METTL14,* or *WTAP* were silenced to maximally 70%, resulting in some residual m⁶A immunostaining (Fig S2A). Scale bar, 10 $\mu$M. **(C)** Comparison of the expression of total mRNA in control growth and following siRNA-mediated knockdown of the "writer" complex (-writers) in HEK-TIA1 cells determined by RNA-Seq. $R^2$ = 0.928, Pearson correlation coefficient. **(D)** Total RNA isolated from the same amount of U2OS-G3BP1 cells grown at permissive (control) conditions or exposed to various AS concentrations and detected with m⁶A antibody or methylene blue. **(E)** Box-plot of m⁶A sites (from the m⁶A-Seq) detected across all mRNAs of untreated cells (control) or exposed to stress (500 $\mu$M AS) and presented as a ratio of the total m⁶A sites (e.g., predicted DRACH motifs designated as A in the ratio m⁶A/A). $P = 2.8 \times 10^{-6}$ control versus stress, Mann–Whitney test.
Source data are available for this figure.

(Figs 1D and S2D) may originate from abundant non-coding RNAs or m⁶A$_m$ modifications. To discriminate only internal m⁶A in mRNAs, we performed global profiling of the RNA methylome (m⁶A-sequencing [m⁶A-Seq] [Meyer et al, 2012; Zhou et al, 2015]) in two independent biological replicates under oxidative stress (500 $\mu$M) and permissive growth. We processed the data for each RNA biotype, which in turn allowed extracting the methylation pattern of mRNAs only. Under control growth conditions, we detected, in total, 8,046 m⁶A peaks at consensus DRACH motifs on 4,488 unique mRNAs. Thus, from all 11,547 cellular mRNAs identified in the RNA-Seq experiment, 38.9% contained at least one m⁶A peak. In response to oxidative stress (500 $\mu$M), the number of m⁶A peaks increased significantly from 8,046 to 9,142 under stress ($P = 2.8 \times 10^{-6}$; Figs 1E and S3A). Also, the number of mRNAs with m⁶A peaks increased (44.2% of 10,791 detected total mRNAs in the RNA-Seq, $P = 2.8 \times 10^{-6}$; Figs 1E and S3A), which supports the notion for stress-induced additional methylation of mRNAs. Therefore, these additional m⁶A peaks appeared not only in non-modified mRNAs, but also on transcripts

that were already partly methylated under control growth (Fig S3A). m⁶A modifications largely overlapped between HEK293 and U2OS cells (Fig S3B), suggesting a conserved methylation pattern among different cell lines.

## mRNAs associated with SGs exhibit a distinct m⁶A pattern

To determine whether SG mRNA clients scored with an enriched methylation pattern, we isolated the SG mRNAs using photo-activatable ribonucleoside cross-linking and immunoprecipitation (PAR-CLIP) (Hafner et al, 2010) (Fig 2A). SGs were stabilized with 4sU-mediated cross-linking of RNAs to the RNA-binding proteins, and thereafter, intact SGs were isolated using established protocols (Jain et al, 2016; Khong et al, 2017). The average size of the isolated SGs (Fig S3C) was similar to the size reported previously (Kim et al, 2006). The SGs contained both mRNAs and proteins (Fig S3C and Table S1) sharing many of previously identified SG clients (Jain et al, 2016; Khong et al, 2017). To select unique mRNA clients segregated in the SGs in response to oxidative stress (500 μM), we used twofold enrichment over PAR-CLIP from the control, unstressed HEK-TIA1 cells and identified 6,020 unique mRNAs associated with the SGs (Fig 2B). Using a 4sU-cross-linking–based approach, we identified a much larger number of SG clients (e.g., 6,020 of 10,791 total mRNAs) than that described in the SG cores (Khong et al, 2017), suggesting that we captured SG clients of the whole SG, including its periphery.

54.7% of these mRNA clients were methylated (Fig 2B). Compared with the control, they had significantly higher proportion of stress-induced methylation sites (Fig 2C) and higher number of methylation sites per transcript (Fig 2D). The stress-induced m⁶A peaks we detected in SG mRNAs (Fig 2C) represent 96% of all mRNAs with increased m⁶A signals in response to oxidative stress (Fig 1E), suggesting that most m⁶A-modified mRNAs reside in SGs. It should be noted that from all predicted m⁶A methylation sites (e.g., DRACH motifs), only a fraction of them were m⁶A modified under permissive growth or oxidative stress exposure (Fig 2D). Although we used anti-TIA1 antibodies to pull down SGs, the identified SG mRNA clients in the PAR-CLIP were enriched not only in TIA1-binding motifs (Fig S3D), implying that through the unspecific 4sU-mediated cross-linking, we captured diverse mRNAs binding to different RNA-binding proteins.

We next analyzed the distribution of m⁶A peaks in different transcript segments of the SG mRNA set, which were binned to equal lengths for comparison (Fig 2E). Following stress exposure, m⁶A peaks markedly increased in the 5′ UTRs and 5′ vicinity of CDSs (Fig 2E and F). In contrast, the m⁶A pattern around the stop codon and the 3′ UTRs (Fig 2E and F) that controls mRNA stability (Meyer et al, 2012; Wang et al, 2014) was unaltered. To test the importance of the m⁶A modifications in the 5′ vicinity of CDSs for the mRNA localization in SGs, we selected a gene, *ARL4C*, which displayed stress-induced increase in the m⁶A level in this region. We introduced the first 102 nt of its CDS in-frame of *YFP* CDS and compared its

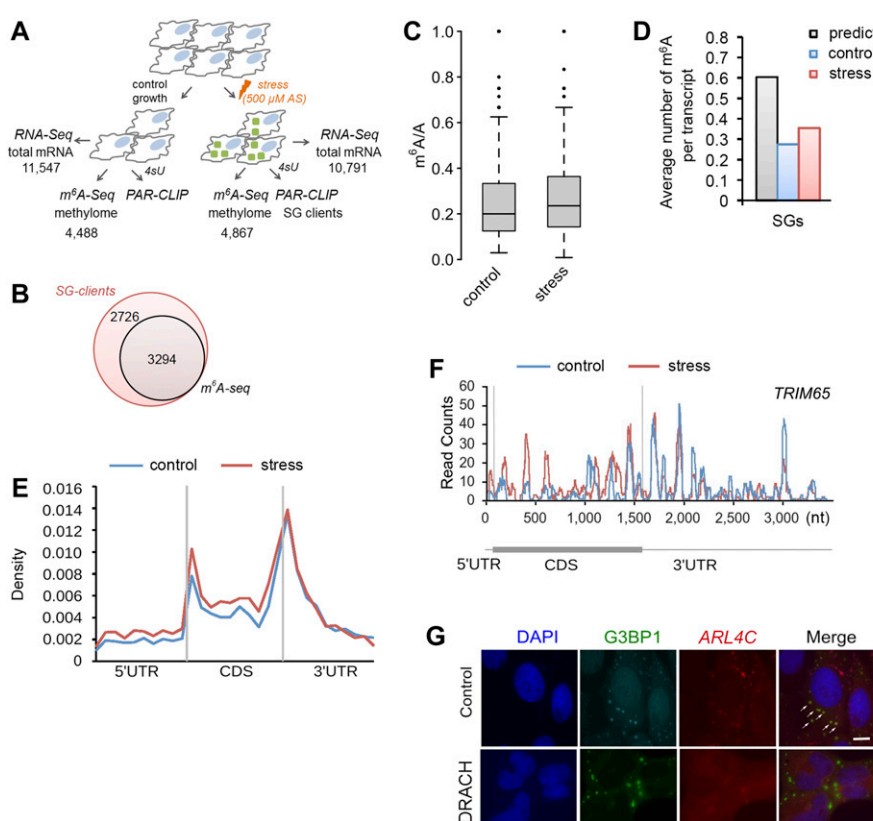

**Figure 2. Site-specific methylation of SG mRNAs in response to oxidative stress.**
**(A)** Overview of the experimental setup. Numbers denote confidently identified mRNAs in each deep-sequencing approach. **(B)** Overlap of the SG transcripts from the PAR-CLIP and m⁶A-Seq data sets. **(C)** Box-plot of m⁶A sites (from the m⁶A-Seq) detected across SG transcripts of untreated HEK-TIA1 cells (control) or exposed to stress (500 μM AS) and presented as a ratio of the total m⁶A sites (e.g., predicted DRACH motifs designated as A in the ratio m⁶A/A). $P = 5.1 \times 10^{-4}$ control versus stress, Mann–Whitney test. **(D)** Average number of m⁶A-modified DRACH motifs detected in the SG mRNAs following stress exposure to 500 μM AS (stress) compared with their methylation level under control growth. $P = 1.49 \times 10^{-5}$ control versus stress, Mann–Whitney test. For comparison, the average number of all putative DRACH motifs (predicted) per transcript is also included. **(E)** Metagene profiles of m⁶A distribution (from the m⁶A-Seq) along different transcript regions of SG mRNAs from untreated (control) or cells exposed to 500 μM AS (stress). $P = 1.4 \times 10^{-3}$ for 5′ UTRs and $P = 1.6 \times 10^{-2}$ for 5′ vicinity of the CDSs; Mann–Whitney test between stress versus control. Transcript regions were binned for comparable lengths. **(F)** An example of stress-induced m⁶A modification in the SG transcript *TRIM65*. **(G)** Deletion of the methylation sites in the 5′ vicinity of *ARL4C* mRNA hinders its localization into SGs. U2OS-G3BP1 cells expressing ARL4C-CFP with unchanged sequence (control) or deleted DRACH motifs (-DRACH) exposed to 500 μM AS. The colocalization of the wild-type *ARL4C-CFP* mRNA with SGs is designated by white arrows. *ARL4C-CFP* mRNA was visualized by in situ hybridization (FISH). SGs (hyperfluorescent loci) were visualized through G3BP1–GFP (green), and nuclei were counterstained with DAPI (blue). 58 of 70 cells (83%) showed loss of colocalization by deleted DRACH motifs. Scale bar, 10 μm. Source data are available for this figure.

localization to a variant in which the methylation sites in this region were removed. The wild-type *ARL4C-CFP* mRNA colocalized in the hyperfluorescent SG loci (Fig 2G). Strikingly, deletion of the methylation sites abolished the colocalization with SG, and the *ARL4C-CFP* mRNA remained diffusively distributed (Fig 2G). Collectively, these data establish a region-selective methylation of the SG following stress exposure.

### Translationally active mRNAs are methylated in the 5′ UTR which does not change in response to oxidative stress

m[6]A in the 5′ UTR selectively regulates translation of transcripts under heat stress in a cap-independent manner (Meyer et al, 2015; Wang et al, 2015; Zhou et al, 2015), which is the prevalent mode of initiation under many stress conditions (Sonenberg & Hinnebusch, 2009). Because we also detected a greater m[6]A level in the 5′ UTR of mRNAs following oxidative stress, we next sought to separately analyze the methylation pattern of translationally active transcripts and those segregated in SGs. Under harsh oxidative stress (500 μM AS), translation was nearly completely inhibited (no apparent polysomal fraction; Fig S1A); thus, we selected a mild stress (200 μM) at which three pools of mRNAs existed in the cytosol: actively translated ribosomes (polysomes), mRNA stalled at translation initiation (80S), and mRNAs sequestered in SGs. To identify the mRNAs in each of these states at 200 μM AS, we applied PAR-CLIP and RNA-Seq along with ribosome profiling (Ribo-Seq) (Fig 3A); in Ribo-Seq, the

ribosome-protected fragments (RPFs) are informative on translating mRNAs (Ingolia et al, 2009). In response to mild stress (200 μM AS), we detected a significant global impairment of translation compared with the control growth (median reduction of the ribosome density [RD] of $\log_2$ = 2.9; Fig 3B), whereas transcription was unaltered (Fig S3E). At mild stress, 2,212 transcripts generated RPFs and these spanned different mRNA expression levels (Fig S3F); most RPFs accumulated were stalled at initiation and early elongation (Fig 3C and D), which is reminiscent of previous observation following thermal stress (Liu et al, 2013). To select genuinely translated mRNAs from the transcripts producing RPFs, we used the translation ratio ($R_t$) as a measure to select mRNAs with a uniform RPF distribution. Following this criterion, 108 genes were selected as translated (Fig 3A and E and Table S2), and the GO terms enriched (enrichment score 12.2) among them were "translation" (fold enrichment 10.26; $P$ = 1.73 × 10⁻¹⁰), "nonsense-mediated mRNA decay" (fold enrichment 20.37; $P$ = 1.43 × 10⁻¹³), and "rRNA processing" (fold enrichment 11.33; $P$ = 2.58 × 10⁻¹⁰). These 108 translated transcripts, which compared with the SG mRNAs are richer in DRACH motifs in their 5′ UTRs (Fig S3G), were mostly methylated under control growth (Fig 3F), which is consistent with previous observations (Meyer et al, 2015; Zhou et al, 2015). Importantly, the m[6]A level in their 5′ UTR did not change following stress exposure (Fig 3F).

Many of the remaining 2,104 transcripts with RPFs displaying halted translation (e.g., $R_t$ > 0.5) were already partitioned in SGs at 200 μM AS or completely segregated in the SGs at harsh oxidative

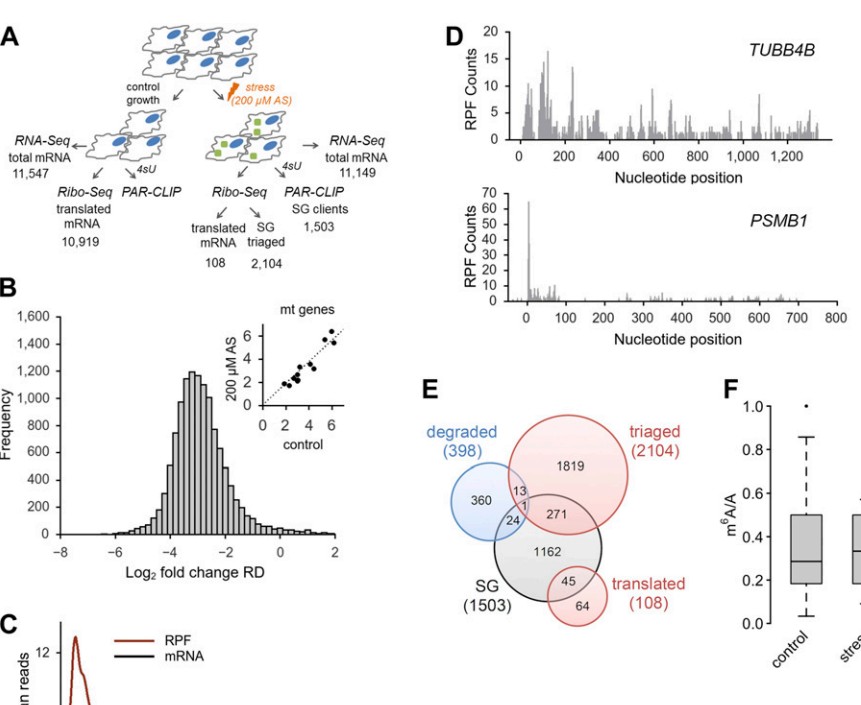

**Figure 3. Oxidative stress globally impairs translation.**
**(A)** Overview of the experimental setup at mild stress (200 μM AS). Numbers denote confidently identified transcripts in each deep-sequencing approach. **(B)** Log-changes of the RD values between control and exposed to 200 μM AS HEK-TIA1 cells. Inset, RD values of the mitochondrially encoded genes which remain unchanged and are used as baseline for normalization of RD values of the nuclearly encoded genes. **(C)** Cumulative metagene profile of the read density as a function of position for both RPF (from Ribo-Seq) and mRNAs (from RNA-Seq) upon exposure to 200 μM AS. The expressed genes were individually normalized, aligned at the start codon, and averaged with equal weight. **(D)** Representative examples of a genuinely translated transcript (*TUBB4B*) and a transcript with stalled translation (*PSMB1*) at 200 μM AS. The first nucleotide of the start codon is designated as zero. **(E)** Venn diagram of the distribution of various transcript groups detected at 200 μM AS. SG, detected in the PAR-CLIP; degraded, identified in the RNA-Seq, red circles, mRNAs with RPFs in the Ribo-Seq. **(F)** Box-plot of m[6]A sites (from the m[6]A-Seq) detected across the actively translated 108 transcripts in control growth or upon stress exposure and presented as a ratio of the total m[6]A sites (e.g., predicted DRACH motifs designated as A in the ratio m[6]A/A). $P$ = 0.97 control versus stress, Mann–Whitney test.
Source data are available for this figure.

stress (500 μM AS; Fig S3H). 69.7% of these mRNAs showed stress-induced enhancement of m6A modifications in their 5′ termini at 500 μM AS. The remaining 30.3% showed no m6A modifications, implying that their triaging into SGs is most likely driven by stress-induced stalling at initiation as described earlier (Buchan & Parker, 2009; Sonenberg & Hinnebusch, 2009; Kedersha et al, 2013). Together, these data indicate that translationally active mRNAs were highly methylated in their 5′ UTRs, but stress did not induce additional m6A modifications. In contrast, the larger fraction of mRNAs triaged to SG displayed stress-induced m6A in their 5′ UTRs and 5′ vicinity of the CDS.

### YTH domain family 3 (YTHDF3) mediates triaging of m6A-modified mRNAs to SGs under oxidative stress

Previous work has identified "reader" proteins as evolutionary conserved cell-type–independent proteins (Edupuganti et al, 2017). These proteins selectively bind the m6A moiety with their YTH domain (Dominissini et al, 2012) and regulate various aspects of RNA homeostasis. To gain a mechanistic understanding of the participation of "reader" proteins in mRNA recruitment to SGs through selective recognition of the m6A sites, we analyzed the distribution of three YTH domain–containing proteins in USO2-G3BP1 cells exposed to 500 μM AS. YTHDF3 colocalized exclusively with the SGs, YTHDF1 exhibited only marginal colocalization with SGs, and YTHDF2 retained its cytoplasmic localization (Fig 4A). At ambient growth conditions, YTHDF1 and YTHDF2 partitioned between the cytosol and nucleus, whereas YTHDF3 resided in the cytosol (Fig S4A), which is in agreement with previous observations (Meyer et al, 2015; Zhou et al, 2015; Li et al, 2017). Importantly, a knockdown of the "writer" complex completely abolished the YTHDF3 localization in SGs but had no noticeable effect on YTHDF1 partitioning into SGs (Fig 4A). Also, siRNA-mediated silencing of

YTHDF3 (Figs 4B and S4B) abrogated the colocalization of the m6A signal with the SGs, suggesting a new role of YTHDF3 "reader" in recruiting m6A-modified mRNAs into SGs. Similarly to the "writer" complex silencing (Fig 1B), the knockdown of YTHDF3 did not change the overall formation of SGs as detected through the SG-scaffold protein G3BP1 (Fig 4B). Importantly, knockdown of the "writer" complex or YTHDF3 markedly decreased the amount of m6A-modified mRNAs in SGs, whereas the amount of non-methylated mRNAs in SGs, detected through the polyA-tag, was influenced to a much lesser extent (Fig 4C); the latter correlates with the observation that 45.3% of all transcripts found in the SGs were not methylated (Fig 2B).

To stratify the specificity of YTHDF3 toward SG substrates, we integrated our m6A-Seq and PAR-CLIP data on SGs with recently published YTHDF3 PAR-CLIP data (Shi et al, 2017). A substantial amount of the genes identified as YTHDF3 clients overlapped with the SG clients (Fig 4D), whereas the overlap with the YTHDF1 and YTHDF2 is much smaller (Fig S4C and D). Together, our results reveal YTHDF3 as a mediator in triaging mRNAs methylated in their 5′ termini to SGs under oxidative stress.

## Discussion

SGs are crucial for facilitating stress response and reprogramming gene expression to maximize cell survival under stress. Our results revealed two modes of triaging mRNAs into SGs following oxidative stress. For the larger fraction of mRNAs (~55%), stress-induced m6A modifications in the 5′ vicinity of the transcripts serve as a specific mechanism for triaging them into SGs (Fig 5). The significance of m6A residues for triaging mRNAs to SGs is further supported by our finding that deletion of DRACH motifs in the 5′ termini of the CDS abrogates the localization of the transcript in SGs (Fig 2G). Another

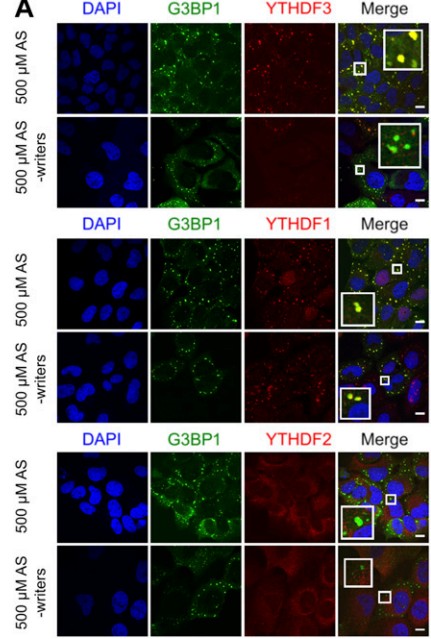

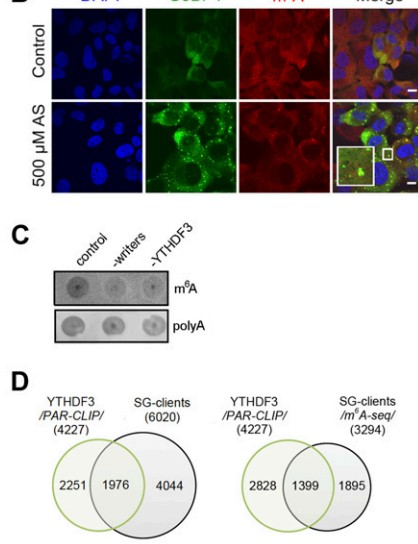

**Figure 4. YDHDF3 colocalizes with m6A-modified mRNA into SGs.**
**(A)** Cellular localization of YTHDF1, YTHDF2, and YTHDF3 in U2OS-G3BP1 cells exposed to oxidative stress (500 μM AS) alone or combined with knockdown (-writers) of the writer complex (*METTL3*, *METTL14*, and *WTAP*). Scale bar, 10 μM. **(B)** siRNA knockdown of *YTHDF3* in U2OS-G3BP1 cells abrogated colocalization of m6A-modified RNA within the SGs. **(C)** Total RNA of SGs isolated from U2OS-G3BP1 with siRNA knockdown of the writer complex (-writers) or *YTHDF3* (-YTHDF3), and control cells exposed to 500 μM AS and detected with m6A antibody or fluorescently labeled oligo-dT primers recognizing the polyA tails of mRNAs. **(D)** Common clients between the YTHDF3 PAR-CLIP target genes (4,227) and total SG clients (6,020 transcripts) and the methylated SG clients detected with high confidence in m6A-Seq (3,294 transcripts). $P = 1.07 \times 10^{-155}$ (for PAR-CLIP, left) and $P = 3.78 \times 10^{-214}$ (for m6A-Seq, right), hypergeometric test.

fraction of mRNAs (~45%), which are not methylated, most likely associate with the SGs triggered by the oxidative stress–induced stalling at initiation (Fig 5) as suggested earlier (Buchan & Parker, 2009; Sonenberg & Hinnebusch, 2009; Kedersha et al, 2013). However, it is possible that stalled ribosomes at initiation and/or early elongation (Fig 3C) would sterically hinder the methylation in these regions. Thus, we may be underestimating the fraction of methylated mRNAs and the stress-induced methylation in the 5′ termini might be more prevalent.

In contrast to the wide belief that m6A modification is static on mRNAs, we found that the 5′ UTR and 5′ vicinity of CDSs methylation are dynamic and induced by oxidative stress. This stress-inducible methylation is recognized by the YTHDF3 reader which relocates those transcripts to SGs (Fig 5). Supportive for the notion that the process is directly mediated by YTHDF3 is our observation that YTHDF3 partitioning in SGs is altered following silencing of methylation, whereas the localization of YTHDF1 and YTHDF2 is unaltered. SGs are enriched with proteins containing IDRs or Gln/Asn-rich prion-like domains with high intrinsic propensity to self-aggregate through hetero- and homotypic interaction (Gilks et al, 2004; Toretsky & Wright, 2014; Lin et al, 2015; Jain et al, 2016). This idea is supported by in vitro observations wherein high concentrations of proteins with IDRs are sufficient to spontaneously form liquid-like droplets (Kato et al, 2012; Lin et al, 2015; Molliex et al, 2015). Structural predictions of the YTH-domain readers with DisEMBL revealed Gln/Asn-rich IDRs in all three proteins, for example, 282–303 aa, 249–299 aa, and 315–351 aa for YTHDF1, YTHDF2, and YTHDF3, respectively. Hence, it is conceivable that the YTH-domain "reader" proteins, all of which are found in the SGs (Table S1 [Jain et al, 2016]), are sequestered in the SGs through unspecific IDR-driven interactions with other SG proteins.

Although we cannot exclude cooperativity among the three YTH-domain readers, clearly only YTHDF3 binds to the stress-induced m6A on mRNAs, and most likely through protein–protein interactions with its IDR relocates them to SGs (Fig 5).

Earlier studies propose that SGs are nucleated by translationally stalled RNAs with assembled initiation factors which serve as scaffolds for RNA-binding proteins (Decker & Parker, 2012; Kedersha et al, 2013). Hence, the primary nucleation of SGs might occur in an m6A-independent manner involving the fraction of mRNAs we detect as non-methylated following stress exposure (Fig 5). A recent study shows that the core SG protein, G3BP1, which nucleates SGs (Kedersha et al, 2016), is repelled by m6A (Edupuganti et al, 2017), and thus might be recruited exclusively to mRNAs lacking m6A modification. Consistent with this model is our observation that silencing of the "writer" complex alters only the association of m6A-modified mRNAs and YTHDF3 with SGs, but not the SG formation in general. Conceivably, the SG nucleation and core formation might occur in an m6A-independent manner involving primarily non-translating mRNAs stalled at initiation, whereas the methylation-driven association of mRNAs might take place in the more dynamic SG periphery (Fig 5).

Our studies show that mRNAs genuinely translated under oxidative stress are enriched in m6A signals in their 5′ UTRs. Unlike the SG clients which are dynamically methylated under stress, the translated pool exhibits high basal methylation (e.g., under permissive growth) which remains unchanged under stress. This raises the intriguing question as to how the YTHDF3 reader discriminates those from SG clients. Under permissive growth, translation of selected transcripts is enhanced by YTHDF1 which binds to select transcripts at m6A in their 3′ UTRs (Wang et al, 2015). YTHDF1 binds

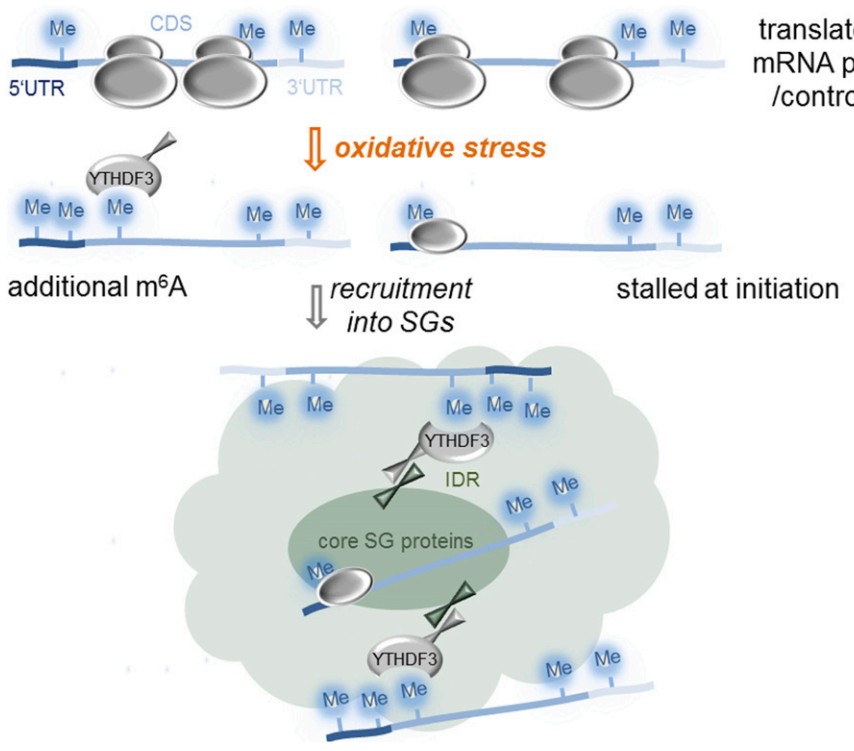

**Figure 5. Proposed model of mRNA triaging into SGs.**
mRNAs associate with SGs either by stress-induced methylation in an YTHDF3-dependent manner (left) or by stress-induced stalling at initiation (right).

simultaneously then to ribosomal proteins of already assembled initiating ribosomes to influence the cap-dependent translation (Li et al, 2017). Although YTHDF3 itself can also associate with ribosomal proteins and m6A-modified 3′ UTRs (Li et al, 2017; Shi et al, 2017), it does not compete but rather facilitates YTHDF1 binding (Li et al, 2017). Our data show that deletion of DRACH motifs in the 5′ termini of the CDS alone is sufficient to abrogate the YTHDF3-mediated localization of the transcript in SGs (Fig 2G) that supports the notion that the 5′ termini of the CDS and not the 5′ UTRs are likely the primary binding site of the YTHDF3 reader. The YTHDF3 binding to 5′ termini in the CDS may then sterically hinder initiated ribosomes to commence elongation. Or, depletion of YTHDF3 from the YTHDF1–YTHDF3–ribosome complex may weaken the YTHDF1 interactions and cause dissociation of the initiation complex. Although the downstream effect is unknown, it is clear that YTHDF3 relocate m6A-modified mRNA in SGs as siRNA-mediated decrease in its concentration abrogates this process. This unexpected feature of YTHDF3 protein in triaging mRNAs to SGs offers a mechanism for dynamic control of the localization of mRNAs during stress.

# Materials and Methods

### Cell culture and siRNA-mediated gene silencing

U2OS cells, stably expressing GFP-tagged SG marker G3BP1, P-body marker DCP1 (Ohn et al, 2008), or HEK293 expressing N-terminally FLAG-tagged TIA1 under doxycycline-dependent promoter (Damgaard & Lykke-Andersen, 2011) were maintained in DMEM, supplemented with 10% FBS, penicillin/streptamycin 250 U, and glutamine 2 mM. For simplicity, cells are called U2OS-G3BP1, U2OS-DCP1, and HEK-TIA1. All cell lines were grown at 37°C in a humidified atmosphere with 5% $CO_2$. Oxidative stress was elicited by adding sodium AS for 30 min at 37°C at 70–80% confluency. Thermal stress was exerted by incubating the cells at 42°C for 2 h. First, 102 nt of the CDS of ARL4C gene were fused in-frame in lieu of the Met codon of CFP. In parallel, three DRACH motifs in the 102-nt ARL4C region were synonymously replaced so that the amino acid sequence remained unaltered. Both constructs were cloned into a pcDNA vector and used for transient transfections.

For siRNA-mediated gene silencing, U2OS-G3BP1 cells were grown to 50% confluency in six-well plates, transfected with 10 mM siRNA unless otherwise stated, dissolved in 4 μl jet prime transfection reagent and 200 μM transfection buffer (Polyplus), and analyzed 48 h after transfection. The siRNA sequences that target METTL3, METTL14, and WTAP were as follows: METTL3, 5′-CUGCAAGUAUGUUCACUAU-GATT-3′; METTL14, 5′-AAGGAUGAGUUAAUAGCUAAATT-3′; WTAP, 5′-GGGCAAGUACACAGAUCUUAATT-3′. Two deoxynucelotides (TT) were added for in-cell stabilization of the oligonucleotides.

### FISH

Two 25-mer DNA oligonucleotides with sequence complementary to the ADAMTS1, ADAMTS3, ADAMTS8, and ADAMTS15 sequences, 5′-AGATAGCGTCCTTCTAGATTTGTGCTGACTGGAGTCACCAGCTCATACTC-3′ and

5′-GAGTATGAGCTGGTGACTCCAGTCAGCACAAATCTAGAAGGACGCTATCT-3′, were annealed in 1× DNA polymerase reaction buffer (Thermo Fisher Scientific) by heating briefly to 95°C and slowly cooled down to room temperature. To 2 μg of annealed oligonucleotides, 1 mM of three unlabeled dNTPs (dGTP, dATP, and dCTP), 0.2 mM Cy5-labeled dTNP, 0.2 U/μl of DNase I, and DNA polymerase I (5–15 U) were added in nuclease-free water (40 μl total volume). Samples were incubated for 15–60 min and purified by standard ethanol precipitation approach.

U2OS-G3BP1 cells grown to 70% confluency were fixed with 10% PFA for 10 min at room temperature and permeabilized overnight at 4°C with 70% ethanol. Cells were rehydrated with rehydration buffer (2× SSC; 300 mM NaCl, 30 mM sodium citrate, pH 7.0, containing 50% formamide) and incubated overnight at 37°C with 30 ng FISH probe dissolved in hybridization buffer (2× SSC containing 10% dextran sulfate, 2 mM vanadyl–ribonucleoside complex, 0.02% RNAse-free BSA, 40 μg Escherichia coli total tRNA, and 50% formamide). Cells were washed twice in rehydration buffer for 30 min and imaged on an Olympus I×81 confocal microscope.

### Polysome fractionation

Cells at 70–80% confluency were pelleted at 850 g and respuspended in polysome lysis buffer (10 mM Tris–HCl, pH 7.4, 5 mM $MgCl_2$, 100 mM KCl, 1% triton X-100, and supplemented with 2 mM DTT and 100 μg/ml cicloheximide). Lysis was performed by sharing 8–9 times through a 21 G needle. 300 μl of lysate was loaded onto 5-ml sucrose gradient (50–15% sucrose in, 50 mM Hepes-KOH, pH 7.4, 50 mM $MgCl_2$, 100 mM KCl, 2 mM cycloheximide, and 2 mM DTT) and separated by ultracentrifugation at 148,900 g (Ti55 rotor; Beckman Coulter) for 1.5 h at 4°.

### PAR-CLIP of SGs

PAR-CLIP was performed following the published protocol (Spitzer et al, 2014) using 4-thiouridine (4sU)–mediated mRNA cross-linking to RNA-binding proteins. Briefly, HEK-TIA1 cells (70% confluent) were supplemented with 4sU to a final concentration of 100 μM and incubated for 16 h before exposure to stress with 200 or 500 μM AS at 37°C for 30 min. Cells were cross-linked with 1,500 μJ/cm2 at 365 nm, washed once with ice-cold PBS, and lysed in lysis buffer (20 mM Tris–HCl, pH 7.4, 15 mM NaCl, 1% NP-40, 0.1% triton-X, and protease inhibitor) by pipetting up and down 8× using a 26 G syringe. The supernatant was cleared at 16,000 g for 10 min and subjected to immunoprecipitation with anti-FLAG antibody-coated magnetic beads. In a parallel procedure, SGs were isolated without 4sU incorporation using cross-linking at 254 nm (1,500 μJ/cm2).

100 μl protein G–coated Dynabeads or Macs protein G micro beads were washed twice with lysis buffer and incubated with 2 μg of anti-FLAG antibody for 1 h at room temperature. The antibody solution was removed and the beads were washed 3× with 900 μl lysis buffer. After incubation with the cell lysate from HEK-TIA1 cells for 2 h at 4°C, the beads were carefully washed twice with washing buffer (20 mM Tris–HCl, pH 7.4, 150 mM NaCl, 1% NP-40, 0.1% triton-X, and protease inhibitor) and directly subjected to RNA extraction for RNA sequencing.

## RNA isolation, RNA-Seq, and Ribo-Seq

RNA was extracted by adding 0.1 volume of 10% SDS, one volume of acidic phenol-chloroform (5:1, pH 4.5) preheated to 65°C and incubated at 65°C for 5 min. The reaction was cooled on ice for 5 min. Phases were separated by centrifugation at 21,000 $g$ for 5 min. Equal volume of acid phenol-chloroform was added to the aqueous phase, separated by centrifugation and supplemented with an equal volume of chloroform:isoamyl alcohol (24:1). Upon separation, the aqueous phase was supplemented with 0.1 vol 3M NaOAc (pH 5.5) and an equal volume of isopropanol. Samples were precipitated for 3 h at −20°C. RNA was pelleted at 21,000 $g$ at 4°C, and the dried pellets resuspended in DEPC-H$_2$O. ERCC RNA Spike-Ins were added upon rRNA depletion using Ribo-Zero Magnetic Gold kit (Illumina) and used to set the detection threshold in each sequencing set. The rRNA-depleted samples were fragmented in alkaline fragmentation buffer (0.5 vol 0.5 M EDTA, 15 vol 100 mM Na$_2$CO$_3$, and 110 vol 100 mM NaHCO$_3$), dephosphorylated and fragments ranging from 20 to 35 nt were size-selected on 15% polyacrylamide gel containing 8M urea. The adapters were ligated directly to the 5′- and 3′-ends as previously described (Guo et al, 2010), converted into cDNA libraries, and sequenced on a HiSeq2000 (Illumina) machine.

Approximately five million HEK 293-TIA1 cells, unstressed or stressed with 200 $\mu$M AS for 30 min, each in two independent biological replicates, were used to isolate mRNA-bound ribosome complexes, followed by extraction of RNase I digested RPFs, as previously described (Guo et al, 2010; Kirchner et al, 2017). Cells were collected by flash-freezing without preincubation with antibiotics, and cDNA libraries from RPFs were prepared with direct ligation of the adapters (Guo et al, 2010; Kirchner et al, 2017) and sequenced on a HiSeq2000 (Illumina) machine.

## m$^6$A-Seq

Total RNA from approximately 45 million HEK 293-TIA1 cells, unstressed or stressed with 500 $\mu$M AS for 30 min, was first isolated using Trizol reagent followed by fragmentation using freshly prepared RNA fragmentation buffer (10 mM Tris–HCl, pH 7.0, and 10 mM ZnCl$_2$). 5 $\mu$g fragmented RNA was saved for RNA-Seq as input control. For m$^6$A-Seq, 400 $\mu$g fragmented RNA was incubated with 10 $\mu$g anti-m$^6$A antibody (#ABE572; Millipore) and 2.5 $\mu$g anti-m$^6$A antibody (#202203; SYnaptic SYstems) in 1× IP buffer (10 mM Tris–HCl, pH 7.4, 150 mM NaCl, and 0.1% Igepal CA-630) for 2 h at 4°C. The m$^6$A-IP mixture was then incubated with Protein A/G beads for additional 2 h at 4°C on a rotating wheel. After washing 3× with IP buffer, bound RNA was eluted using 100 $\mu$l elution buffer (6.7 mM m$^6$A 5′-monophosphate sodium salt in 1× IP buffer), followed by ethanol precipitation. Precipitated RNA was used for cDNA library construction and high-throughput sequencing described below.

## Preprocessing of the sequencing reads

Sequencing reads were trimmed using fastx-toolkit (quality threshold: 20), adapters were cut using *cutadapt* (minimal overlap: 1 nt), and processed reads were mapped to the human genome (GRCh37) using Bowtie either uniquely or allowing multimapping with a maximum of two mismatches (parameter settings: -l 16 -n 1 -e

50 -m 1 or 10 −strata −best y). Uniquely mapped RPF reads (Ribo-Seq) or fragmented RNA reads (RNA-Seq) were normalized as reads per million mapped reads or reads per kilobase per million mapped reads. All sequencing reactions were performed in biological replicates. Based on the high correlation between the replicates ($R^2$ > 0.9 for all data sets, Person coefficient), reads from biological replicates were merged into metagene sets (Ingolia et al, 2009).

m$^6$A-Seq and input RNA-Seq reads (20–40 nt) were aligned to NCBI RefSeq mRNA sequences and UCSC genome sequences (hg19 for human) using Tophat (−bowtie1 −no-novel-juncs -G) as described previously (Trapnell et al, 2009).

## Data set processing

Under control growth conditions most of the transcribed mRNAs were also translated in HEK-TIA1 cells (Fig S4F). The RD for each transcript (previously defined as "translation efficiency" [Ingolia et al, 2009]), was computed as follows:

$$RD = \frac{RPF\ [RPM]}{mRNA\ [RPM]} \tag{1}$$

RD values of all protein-coding genes were normalized to the RD of mitochondrial genes as described in Iwasaki et al (2016). Mitochondrial genes were used for normalization as their expression under stress remained unchanged.

Cumulative profiles of the read density for RPFs and mRNA have been computed as described in Gerashchenko et al (2012). High ribosome occupancy at the start of the CDS following exposure to oxidative stress indicated that not all RPFs reported translation (Fig 3C). To distinguish between genuinely translated transcripts and those whose translation was inhibited by stress, we set the following ratio $R_t$:

$$R_t = \frac{Total\ RPF\ reads\ of\ initial\ stalled\ peak\ (first\ 100\ nt)\ [RPKM]}{Total\ RPF\ reads\ over\ the\ full\ gene\ [RPKM]} \tag{2}$$

At 200 $\mu$M AS, 108 mRNAs exhibited $R_t \leq 0.5$ and were considered as actively translated, whereas for 2,104 genes, we detected RPFs largely stalled at initiation with $R_t > 0.5$ and designated them as triaged for SG.

In the PAR-CLIP experiments, SG clients in cells stressed with 200 or 500 $\mu$M AS were selected using a threshold of log$_2$ = 2 over control growth. The variability between biological replicates in the PAR-CLIP experiments (Pearson correlation coefficient) from cells exposed to 200 or 500 $\mu$M AS was $R^2$ = 0.695 and 0.735, respectively. Furthermore, the correlation between the selected SG clients at both stress conditions was very high (Fig S3G). The data set at 200 $\mu$M AS comprises PAR-CLIP detected and triaged in the Ribo-Seq (Fig 3D). Most of the transcripts identified in the Ribo-Seq with halted translation and designated as triaged for SG were also found among the SG clients at harsh stress (500 $\mu$M AS). Thus, all selected mRNAs (either in the PAR-CLIP data sets or designated as triaged in the Ribo-Seq) were merged together into a metagene set of SG clients containing, in total, 6,020 transcripts. These mRNAs found in the SGs span a large expression range (Fig S3F). Statistical analysis was mainly performed in R and SigmaPlot (Systat Software).

## Motif analysis

De novo search for DRACH motifs was performed using FIMO (FIMO-MEME suite; http://meme-suite.org/doc/fimo.html) and the threshold was at $P < 0.001$. The corresponding transcript groups were prepared with Ensembl Biomart. For comparing the number of DRACH motifs in each transcript region, 5′ UTRs, CDSs, and 3′ UTRs were divided into equal bins of comparable length, and the amount of motifs in each segment was averaged over the whole gene set in the selected group. A general motif search among the SG clients was performed using a MEME suite. Gene function analysis (GO enrichment) was performed with the DAVID tool.

## Identification of the m⁶A sites

All full-length mapped reads were used to generate an $m^6A$-Seq coverage profile for individual genes. To compare metagene $m^6A$ profiles between control and stress (500 $\mu$M AS) samples, the raw coverage values were first internally normalized by the mean coverage of each individual gene (for genes with multiple mRNA isoforms, the longest isoform was selected). The genes with maximal coverage value less than 15 were excluded from further consideration. The normalized $m^6A$-Seq profiles of the individual gene were next subtracted by corresponding RNA-Seq profile to generate an adjusted $m^6A$-Seq profile. The metagene profile used for between-sample comparison (control versus stress) was finally derived by averaging all adjusted profiles of individual genes.

The identified $m^6A$ peaks in the $m^6A$-Seq were assigned to the predicted DRACH motifs. Peaks occurring in regions that cover at least one predicted DRACH motif were selected for further analysis. If more than one DRACH motif was found within an $m^6A$ peak, all of them have been considered as methylated. Metagene profiles of the distribution of the $m^6A$ sites along different transcript segments (Fig 2F) were performed by determining the ratio between $m^6A$-modified DRACH motifs detected in $m^6A$-Seq and total number of predicted DRACH motifs in each transcript segment. To compare the $m^6A$ peaks in HEK-TIA1 to those of U2OS-cells from a previously published $m^6A$-Seq data set (Xiang et al, 2017), the $m^6A$-modified DRACH motifs identified in HEK-TIA1 were compared with those in U2OS (Fig S3B).

## Data set availability

Deep sequencing data from RNA-Seq, Ribo-Seq, PAR-CLIP, and $m^6A$-Seq were deposited in the BioSample data base (https://www.ncbi.nlm.nih.gov/biosample/) under accession number SRP121376.

## Supplementary Information

## Acknowledgements

We are grateful to Dr. Hartmut Schlüter and Parnian Kiani from the mass spectrometry core facility University Medical Center Hamburg-Eppendorf, Hamburg, for the help with mass spectrometry analysis. We thank Lindsey Bultema for the transmission electron microscopy images. This work was supported by grants to Z Ignatova from the Deutsche Forschungsgemeinschaft (IG73/14-1 and IG73/14-2).

## Author Contributions

Z Ignatova: conceptualization, formal analysis, supervision, funding acquisition, project administration, and writing—original draft, review, and editing.
M Anders: investigation.
Chelysheva: data curation, formal analysis, investigation, and writing—original draft.
I Goebel: investigation.
T Trenkner: investigation.
J Zhou: investigation.
Y Mao: investigation.
S Verzini: methodology.
S-B Qian: writing—review and editing.

## Conflict of Interest Statement

The authors declare that they have no conflict of interest.

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
