## [Reviewer comments · Life Science Alliance]

Dynamic m⁶A methylation facilitates mRNA triaging to stress granules

Maximilian Anders, Irina Chelysheva, Ingrid Goebel, Timo Trenkner, Jun Zhou, Yuanhui Mao, Silvia Verzini, Shu-Bing Qian, and Zoya Ignatova
DOI: 10.26508/lsa.201700113

Review timeline:	Submission date:	21 June 2018
	Editorial Decision:	21 June 2018
	Revision received:	22 June 2018
	Accepted:	25 June 2018

Please note that the manuscript was previously reviewed at another journal and the reports were taken into account in inviting a revision for publication at *Life Science Alliance* prior to submission to *Life Science Alliance*.

Report:

(Note: Letters and reports are not edited. The original formatting of letters and referee reports may not be reflected in this compilation.)

1st Editorial Decision

21 June 2018

Thank you for transferring your manuscript entitled "Dynamic m⁶A methylation facilitates mRNA triaging to stress granules" to Life Science Alliance. The manuscript was assessed twice by expert reviewers at another journal, and the editor confidentially transferred these reports to us with your permission.

One of the reviewers supported publication of your work, while another reviewer still noted some issues and didn't think that the insight offered goes far enough at this stage. The latter concern is not precluding publication here, and we would therefore like to invite you to revise your work for publication in Life Science Alliance. Based on the reports at hand, we would expect a point-by-point response to the individual concerns raised and accordingly text changes. Importantly, we would like to ask you to discuss differently the role of YTHDF3 as well as the role of METTL3 in the cytoplasm (in light of ref#1's comments). Please also add the requested quantification for the co-localization analysis.

TRANSFERRED REFEREE REPORTS ON REVISED VERSION

Referee #1:

Comments for Anders et al.,
RNA modification at the N⁶ position of adenosines (m⁶A) is essential for viability of mice and plants. It is implicated in a variety of RNA processing events ranging from splicing, polyadenylation site usage to export, mRNA decay and translation. The methylation event is believed to be a co-transcriptional process as the methyltransferase METTL3 is nuclear. The modification is proposed to act as a dynamic mechanism by which the cell can tag transcripts emerging from the transcription machinery depending on cellular stress conditions like heat shock, UV irradiation or conditions that reduce transcription elongation. Furthermore, studies have shown that there is almost no difference in m⁶A methylation on transcripts associated to the chromatin and those found in the cytoplasm, under normal conditions (meaning the mark is added in the nucleus and not much of demethylation

or methylation happens in the cytoplasm).

Anders et al addressed the role of m6A in moving mRNAs into Stress Granules (SGs) under stress induced by arsenite. They describe increased m6A methylation at the 5' end of transcript coding region and propose that it helps to triage mRNAs into SGs during stress. They implicate a role for an m6A reader protein YTHDF3 in selectively recognizing these 5'UTR marks.

Major concerns

1. Fig EV2B. Partitioning of METTL3 to the cytoplasm, but not of METTL14 etc. The suggestion that METTL3 alone is now going to be active in the cytoplasm is not possible based on structural and available biochemical data from the literature. Purified METTL3 alone is inactive. METTL14 is required to maintain the catalytic pocket by interacting with the "interface loop" of METTL3 (see the three structure papers). So the subsequent claim that under stress increased methylation is due to cytoplasmic methylation by mislocalized METTL3 is not supported. A simple way to rule this out is by blocking transcription. It is already known that increased m6A methylation can take place co-transcriptionally (Slobodin et al., 2017). The authors might also try labelling transcripts (SLAM-seq) at different time-points and follow their methylation profile.
2. Figure 2G: Molecular basis for targeting RNAs to SGs. This is the central question posed by the authors. The authors describe an increase in m6A at the 5'UTR and 5' vicinity of CDS. The experiment in Fig2G shows that a reporter carrying 102 nt of the CDS from ARL4C (a SG-enriched transcript) is enough to target the reporter to SG. This fusion of 102 nt was made to the coding region of the reporter, which now is extended at the N-terminus. Mutation of three DRACH motifs within this 102 nt stretch abolished localization. This shows that just three DRACH motifs in the N-terminus of the coding sequence is sufficient for targeting. How general is this observation? How many cells show this co-localization pattern? Are these three sites methylated in the context of the reporter? Will presence of just three additional motifs (not the entire 102 nt) in fusion to the reporter be sufficient? I am trying to understand the nature of the signal for triaging to the SG? What about m6A on the 5'UTR? This experiment may be exploited further to understand the molecular basis for this observation. How to explain SG localization of RNAs that are not methylated? Or SG localization of those RNAs which are methylated?
3. Figure 4: Role of YTHDF3 in triaging mRNAs into SG. It is hard for this reviewer to comprehend how it can carry out this function of recognizing m6A on the 5' CDS regions. How is it different from the other YTH proteins? There is accumulated evidence from a number of studies that the three YTHDF proteins (YTHDF1-3) are in fact largely redundant, and share the same binding sites across the transcriptome (Patil et al., 2018).

Minor

1. Figure legends 2G: "Deletion of the methylation sites in the 5' vicinity of ARL4C mRNA renders its localization into SGs. Renders should be hinders?"
2. Somewhere else in the text "Motives" should be motifs.
3. Figure 2G: The reporter assay. The text in page 7 mentions YFP fusion, while legends and methods mentions CFP. Correct it.

Referee #2:

The authors have performed additional experiments to strengthen their findings and conclusions. Their new information and revised manuscript have answered most of my previous questions with regard to stress-inducible m6A. While no mechanistic insight for YTHDF3 is provided, I think the concept of m6A facilitated mRNA triaging is better demonstrated in this version of manuscript. Thus, I recommend the publication of this paper.

TRANSFERRED REFEREE REPORTS AND AUTHOR RESPONSE ON INITIAL VERSION

Referee #1:

The manuscript by Anders et al., proposes a role for dynamic m6A RNA methylation in sorting transcripts into stress granules (SG) during cellular stress. Stress conditions result in a global shutdown of translation and removal of transcripts stalled in translation initiation to SGs. This accumulation is helped by interaction of the SG components with such RNAs, and also by protein-protein interaction of SG-localizing

proteins via their intrinsically disordered regions (IDRs). However, some transcripts continue to be translated.

Presence of m⁶A at 5'UTRs is shown to promote cap-independent translation during stress (Meyer et al., 2015, Coots et al., 2017) and m⁶A at 5' UTR also allows alternative translation (affects ribosome scanning and alternative start codon selection) during stress response (Zhou et al., 2018).

Anders et al used arsenite to induce oxidative stress and make several observations.

1. They identify an increase in m⁶A RNA methylation under stress.
2. The de novo methylation is preferentially on SG-targeted mRNAs (translationally-repressed).
3. The pattern of methylation is distinct from those of transcripts which are translationally active, as it is more on 5'UTR and near start of coding sequencing.
4. They implicate the m⁶A reader YTHDF3 in this process of sorting such methylated RNAs to the SG.

Comments

1. The term "de novo methylation" is very misleading. The authors see an increase in m⁶A levels upon stress (Figure 1D). I don't understand the meaning of de novo in this context. Every m⁶A RNA methylation is de novo (in non-stressed and stressed cells) as there is no template methylation to serve as a guide (similar to maintenance DNA methylation).

We use the term 'de novo' to describe site-selective m⁶A modifications that are induced on mRNAs following stress exposure. As we show (Fig. 2B, 2D, 2F) many mRNAs are naturally m⁶A modified under permissive growth, in particular around the stop codon and their 3'UTRs. This methylation pattern is known to control mRNA stability and is consistent with earlier observations (Meyer et al, 2012; Wang et al, 2014). Following stress exposure, m⁶A peaks markedly increased in the 5'UTRs and 5' vicinity of CDSs (Fig 2D and F). Mostly, the number of methylation sites per transcript increased suggesting that stress-induced new m⁶A modifications were introduced primarily on already methylated transcripts (discussed on p.6 in the manuscript). To those newly introduced methylation sites we refer to as 'de novo' sites, as they occur only under stress exposure. Reflecting on the comment of the Reviewer we consider that without explicit definition of what is defined as 'de novo' the usage of this term might be misleading and indeed might be taken as a process occurring at the birth of mRNA. Thus, we offer to rephrase this in the text to 'stress-induced new methylation sites' or 'stress-induced additional methylation sites'. Furthermore, using the term sites instead of transcripts also clarifies that only positions within the transcript are meant by not whole transcripts.

2. Figure EV2C. Presence of METTL3 in the cytoplasm under stress. The authors state that cytoplasmic METTL3 could be responsible for de novo RNA methylation. There is no experiment that rules out the possibility that the increased methylation seen during stress conditions is indeed not taking place in the nucleus. Ideally such experiments should be conducted under conditions where transcription is shutoff, so that one can distinguish between co-transcriptional RNA methylation vs cytoplasmic methylation of already transcribed RNAs, upon stress.

We agree with the Reviewer that the analysis cannot discriminate between m⁶A modification taking place in the cytosol and nucleus. Since we do not analyze the redistribution or partitioning of mRNAs between nucleus and cytosol, but only determine the m⁶A sites per transcript independent of the cell compartment it takes place, this analysis is not central to our story. However, we felt that we should report the observation about METTL3 partitioning in the cytosol upon stress exposure, since it resembles recent observations in human cancer cells and mouse embryonic stem cells (Alarcon et al, 2015b; Lin et al, 2016). Thus, we have put result in the extended data (Fig. EV2C) and very carefully phrased it as a possibility:

p.5: '...implying that stress-induced changes in m⁶A modifications on mRNA could take place both in the nucleus but also directly in the cytosol.'

3. Figure 2B-C: The increase in m⁶A sites on transcripts is very mild at best.

We agree with the Reviewer that when presenting the data as m⁶A sites counts on the background of all possible DRACH motifs, despite being significant, the numbers appear small (Fig. 2B, C). Moreover, most likely the signal is dominated by the m⁶A sites around the stop codon, which are abundantly methylated also under permissive growth. Hence, we use another comparison which depicts the m⁶A sites in position-dependent manner (Fig. 2D, F) which shows a significant increase within the 5'UTRs and 5'vicinity of CDSs with no changes within the naturally highly methylated 3'UTRs. These types of analyses are established in the literature to analyze m⁶A-Seq data (Zhou et al Nature 2017).

We wish to emphasize that in all those representations the sites are normalized or weighted equally, as commonly established in the literature for transcription-wide analysis, to exclude a dominating effect of few transcript or few methylation site of a transcript (see for comparison how unweighted intensity from a the single gene presentation would look like (Fig 2D)). Furthermore, it should be noted that the changes we report are of the same order of magnitude what has been reported for changes in m⁶A pattern under other conditions (Zhou et al, Mol Cell 2018, Fig. 6A and 7A reporting on starvation stress; Zhang et al, Nature 2017; Fig. 1C representing changes upon abrogated methylation; Lin et al, 2017, Cell Res.; Fig. 4B; spermatogenesis in differentiation process; Coots et al, Mol Cell 2107; Fig. 3D reporting on METTL3 sensitivity).

4. Figure 2E: 54% of all SG transcripts are methylated. So how do the remaining 46% of SG transcripts get to the SG in the absence of m⁶A RNA methylation.

Possible mechanisms of selection of (additionally) methylated (54%) vs (not additionally) non-methylated (46%) mRNAs under stress we present in the discussion (p. 10). We propose the following which also considers current models from the literature:

“Our results demonstrate a prevalent stress-induced methylation in the 5'UTR and 5'vicinity of the CDSs, however not all cellular mRNAs we detected in the SGs are methylated (Fig 2E). Since oxidative stress induces stalling of ribosomes at initiation and/or early elongation (Fig. 2C), it is possible that ribosomes would sterically hinder the methylation in these regions. SGs are nucleated by translationally-stalled RNAs with assembled initiation factors which serve as scaffolds for RNA-binding proteins (Decker & Parker, 2012; Kedersha et al, 2013). Hence, the primary nucleation of SGs might occur in m⁶A-independent manner. Recent study shows that the core SG protein, G3BP1, is repelled by m⁶A (Edupuganti et al, 2017). It is conceivable that G3BP1 is recruited exclusively to mRNAs lacking m⁶A modification. Consistent with this model is our observation that silencing of the ‘writer’ complex alters only the association of m⁶A-modified mRNA and YTHDF3 with SGs, but not their size and number (Fig 1B and 4A).”

Reflecting on the comment of the Reviewer, we feel that we should modify our model (Fig. 4D) and expand it by an additional branch to illustrate the stalling at initiation as alternative possibility of recruiting mRNAs into stress granules.

5. Figure 2D-F: The central claim of this paper: increase in m⁶A methylation at 5'UTR and CDS. Can the authors demonstrate that at least some of the SG transcripts where the 5'UTR or CDS methylation sites are mutated, will now fail to go to SG? Additional support using mutation analysis of reporters is needed. Reporters with RRACH motifs at 5' UTR or with mutated may be verified for their accumulation in SG in response to stress. The whole point about quantitative increase in methylation is hard to appreciate from the provided data.

In the revised version we are willing to include such data, which support the idea that methylation of a large portion of transcripts is a (see the brief summary below). We thank the Reviewer for rising this point as addition of these data will strengthen the paper's message.

Briefly, from the methylated SG clients we selected one gene (*ARL4C*) which is additionally methylated in the 5' vicinity following stress exposure. We introduced the first 102 nt of its coding sequence in-frame, upstream of the YFP CDS (named wild-type construct). In parallel, the DRACH (formerly RRACH) motifs in this 102nt-region were silently mutated

(named –DRACH construct). Strikingly, while the wild-type mRNA colocalizes with stress granules, the colocalization with stress granules is abolished in the –DRACH construct with deleted methylation sites in the 5' vicinity.

Fig. R1. U2OS-G3BP1 cells expressing ARL4C102-CFP with unchanged sequences (control) or with deleted DRACH motifs (-DRACH). Cells were exposed to harsh (500 μ M AS) oxidative stress for 30 min and ARL4C-CFP mRNA was visualized by in situ hybridization (FISH). SG (hyperfluorescent loci) were visualized through G3BP1-GFP (green), nuclei were counterstained with DAPI (blue). Scale bar, 10 μ m.

6. Figure 3: It is not clear to this reviewer how translated mRNAs still have higher m6A methylation (Figure 3E), but SG mRNAs that also have higher m6A methylation fail to get translated. These experiments suffer from the fact that there is no clarity about transcriptional effects upon stress induction vs post-transcriptional stability of the transcripts. How does the cell distinguish the two species of highly methylated transcripts: one is allowed to translated, while the other is kept in SG repressed?

We believe that this comment is based on some misinterpretation of the data. Specifically, this figure depicts a comparison of the methylation of the 108 genes translated under stress (see the legend to the figure) and statistical analysis reveal that the changes are not significant (see the p values from the Mann-Whitney test included in the figure caption). When describing these data we also state in the text the following on p. 6: “Importantly, the m⁶A level of the translated genes did not change between control and stress (Fig 3E), implying no significant contribution of *de novo* methylation for selecting the translated mRNA pool. In marked contrast, under stress the m⁶A signals increased in the 5'UTRs of the SG mRNA clients (Fig 2F).” This figure represents no significant changes.”

Furthermore, the Reviewer mentions that we do not consider the effect on transcription level and post-transcriptional effects on stability. We kindly disagree and wish to emphasize that we perform the RNA-Seq to address exactly those effects. We made following observations:

- A. The used dose of stress does not trigger global transcriptional response and alter mRNA stability only marginally. On p. 4, including Fig. EV1C we state: “The maximal stress dose (500 μ M AS) and exposure (30 min) we used, caused only marginal changes in the total mRNA levels as revealed by RNA-Seq (Fig. EV1C). Overall, comparing to the total mRNAs detected under permissive growth (reads per kilobase per million reads (RPKM) over the spike-in threshold), we did detect only 6.5% decrease of the total mRNA under stress (Fig EV1C), consistent with previous observations that short AS exposure does not trigger a global transcriptional response or alter the stability of few specific mRNAs (Andreev et al, 2015).”
- B. Silencing of the writer complex does not globally change the mRNA abundance. On p. 4 including Fig. 1C we state: “Since m⁶A modulates mRNA stability (Mauer et al, 2017; Wang et al, 2014), we next determined the global effect of silenced ‘writer’ complex on the mRNA abundance using RNA-Seq. Overall, comparing to the total mRNAs detected under permissive growth, we did not detect substantial changes in the global mRNA levels following knockdown of the ‘writer’ complex (Fig 1C).”
- C. For mild stress (200 μ M AS) on p. 7, including fig. EV3E we state: “In response to mild stress (200 μ M AS), we detected a significant global impairment of translation

compared to the control growth (median reduction of the ribosome density of $\log_2=2.9$; Fig 3B), while transcription was unaltered (Fig EV3E).”

7. Figure 4: Why is YTHDF3 able to carry out such sorting of transcripts to SG? What makes it different from YTHDF2 and YTHDF1? If recognition of RNAs via m6A is by YTH domains, that is not going to discriminate m6A marks at 5' UTRs vs 3'UTR (as depicted in the model Fig 4D). Will tethering of YTHDF3 to a transcript lead to its accumulation in the SG? Will the N-terminal disordered region of YTHDF3 is sufficient to this targeting to SG, when tethered to a transcript? There is no explanation why YTHDF3 can specifically recognize SG RNAs, when many more (46%) SG RNAs are not methylated, hence not linked to YTHDF3 for targeting to SG.

Our knowledge on the recently discovered family of the ‘reader’ YTH-domain proteins is constantly growing: published literature along with our study clearly show a functional ‘specialization’ of three readers, with YTHDF1 specifically binding to the 5'UTRs (Li et al, 2017; Shi et al, 2017), YTHDF2 specifically targeting the 3'UTRs and triaging mRNAs for degradation (Meyer et al. 2012; Wang et al, 2014), and YTHDF3 being active under stress response (this study). While this functional specialization is recognized in the literature, the mechanism of recognition/discrimination for any reader is lacking. Many laboratories, including ours are eagerly looking to dissect this, but the difficulty stems from the fact that the three ‘readers’ exhibit different concentration in the cell and manipulation of the cellular concentration of any of them (including deleting domains to elucidate specific function) causes disbalance in the entire ‘reader’ network and triggers some unspecific binding and overlap between their clients (Shi et al, 2017). In this study we focused on elucidating the reader specificity under stress. In the Discussion (p.10) we provide some discussion on the binding specificity and the link between the readers.

In the revised version we will modify our model to include both possible scenarios of recruiting mRNA into SGs, e.g. through YTHDF3-mediated m6A recruitment and by the stalling at initiation (see the explanation to p. 4).

Minor comments

1. The word "dynamic" methylation in the title is confusing to the field, as it refers to reversibility of m6A modification. Here it is at best transcriptional increase of methylation, as shown previously for UV damaged transcription units.

We agree with the Reviewer that we do study only one process – the methylation and not its reversibility following stress relief, using dynamic might be misleading. Thus, we would change the title to “m⁶A modification triages mRNA to stress granules”

Referee #2:

In this study by Anders et al, the authors report the finding that dynamically induced m6A triages mRNA into stress granules. More specifically, under oxidative stress, a subset of mRNA are de-novo m6A modified at their 5'-UTR and CDS. Such de novo methylation enables these mRNA to be triaged into SG, via the m6A reader YTHDF3. The finding hence reports a novel function of dynamic m6A in escorting selected mRNA transcripts to SG.

This is a novel and timely study that reports a new physiological role of m6A under oxidative stress. In particular, this demonstrates the concept of "dynamic" modification in orchestrating important cellular functions. Since the dynamics of m6A appears to undergo some sort of debate recently, this study provides a convincing story and should be published for the community. Nevertheless, I do have several comments below before I see this manuscript fully ready for EMBO J.

We are pleased to read this general comment of the Reviewer!

Major points:

1. Why the increase of m6A appears to be so different (a lot increase in Fig. 1D but not

only mildly in 1E)? Since total RNA is used in 1D, does it mean in fact there are more de novo m6A sites in polyA minus fraction?

Fig. 1D represents changes in methylation of total RNA, which includes all RNA entities (mRNA, rRNA, tRNA, ncRNA...). As we explain in the manuscript on p. 5, the m⁶A antibodies do not discriminate between different types of methylation (m⁶A vs m⁷G cap is 2'-O-methyladenosine (A_m) which can be further methylated at N⁶ position (m⁶A_m)) and between single RNA entities and the signal is most likely dominated by rRNAs. Thus, we performed m⁶A-seq and by data analysis we select only mRNAs which are represented in all other plots including Fig. 1E. The methylation on mRNA is indeed much lower than methylation of non-coding RNAs which is also well represented in the literature.

The explanation in the text we offer to this is:

“The first nucleotide in the m⁷G cap is 2'-O-methyladenosine (A_m) which can be further methylated at N⁶ position (m⁶A_m) (Mauer et al, 2017), which can be also targeted by the m⁶A-antibodies. Furthermore, a large fraction of non-coding RNAs and rRNA from the small subunit within the initiation complex found in the SGs (Decker & Parker, 2012; Kedersha et al, 2013; Khong et al, 2017) can also be methylated and recognized by the m⁶A-antibodies. To discriminate internal m⁶A in mRNAs only, we performed global profiling of the RNA methylome in two independent biological replicates (m⁶A-Seq; (Meyer et al, 2012; Zhou et al, 2015)) under oxidative stress (500 μM) and compared the intrinsic m⁶A sites in mRNAs to those under permissive growth.”

2. The authors performed m6A-seq under conditions harsh OS and compared that with normal conditions. Why not perform m6A-seq for the mild stress as well? Because the authors themselves mentioned in later section that "some of these partitioned in SG at 200 μM AS or were completely segregated in the SGs at harsh oxidative stress". An m6A-seq (two replicates would be enough) under the mild OS condition would enable the identification of perhaps the more sensitive de novo m6A and hence reveal perhaps a dosage-dependent layer of dynamic modification in regulation. This experiment could potentially significantly improve the story.

The Reviewer suggests an additional m⁶A-Seq at mild stress condition. While we understand the reasoning of the Reviewer, given the resolution restriction of this method we doubt that it may provide additional information. Our argument has rather technical character and explains why we initially decided to use only two very distinct conditions, namely control and harsh stress. The peak calling in m⁶A-experiment includes two steps (see also the description in Methods section): (1) Peak-Over-Median score (POM) is derived by calculating the ratio of mean read coverage in the window to the median read coverage of the whole gene, which is used as internal threshold within m⁶A-seq. (2) Peak-Over-Input (POI) score is extracted after subtraction of the corresponding RNA-seq sample which is used as a background to detect only the m⁶A specific enrichments. We are afraid, that under the mild stress, even if the obtained coverage in the potential regions will be higher than in control sample the values still may not pass the filter of selection to be considered as a peak. However the selection criteria cannot be weakened to avoid the false positives. The number of transcripts that will remain under normalization and subtraction will be much smaller and we are afraid that their amount would not have an intermediate number (between control and 500 μM), but rather be very small over the control which will constrain the statistical analysis.

The Reviewer expects that m⁶A-Seq at mild stress would identify transcripts that partition at 200 μM between cytosol and stress granules and are completely segregated in stress granules following harsh stress. This separation in two pools further decreases the sensitivity. With this reasoning, we initially decided to use ribosome profiling and PAR-CLIP at the intermediate stress, each of which has much higher sensitivity. The combined analysis of both approaches allows us to identify SGs that in the stress granules already and those whose translation is stalled at initiation and are triaged to stress granules.

3. In Fig 2E, the authors report 54.7% overlap of SG transcripts and m6A modification. However, this overlap does not distinguish the de novo m6A modification that occurs after oxidation stress (OS) from the pre-existing ones. IN particular, the authors suggest that "de novo m6A modification following stress exposure likely occurs primarily on already

methylated transcripts", hence making it difficult to distinguish the role between the de novo m6A sites and the pre-existing ones. How do we know that the pre-existing m6A (presumably also bound by YTHDF3) does not play a role in triaging mRNA to SG? One experiment to look into for this question is to provide the percentage and identity of the de novo methylated sites and transcripts after OS.

We thank the Reviewer for this suggestion and intend to include the analysis in the revised manuscript version. As mentioned above (Ref.#1, comment 1), following stress the number of methylation sites per transcript increases suggesting that stress-induced new m⁶A modifications were introduced primarily on already methylated transcripts (discussed on p.6 in the manuscript). We calculated all newly introduced methylation sites under stress and from the methylated SG clients 39% of the transcripts exhibit at least one newly added methylation site.

4. The finding that YTHDF3 co-localizes with SG is very interesting. However, there is little mechanism with regard to this. For instance, what domain (the authors themselves discussed QN-rich domains of the YTH family proteins) of YTHDF3 is important for this co-localization? What determines the specificity of YTHDF3? And why is Y3, but not other YTH proteins involved in triage? The comparison between m6A and PAR-clip data on SGs with PAR-clip data of YTHDF3 that is obtained under normal cellular condition (but not under oxidation stress used in this study, which is shown by the authors to introduce a re-localization of YTHDF3) barely provides the bona fide targets of Y3 under OS.

We find also fascinating that already published literature along with our study clearly show a functional 'specialization' of three readers, with YTHDF1 specifically binding to the 5'UTRs (Li et al, 2017; Shi et al, 2017), YTHDF2 specifically targeting the 3'UTRs and triaging mRNAs for degradation (Meyer et al. 2012; Wang et al, 2014), and YTHDF3 being active under stress response (this study). While this functional specialization is recognized in the literature, the mechanism of recognition/discrimination for any reader is lacking. Many laboratories, including ours are eagerly looking to dissect this, but the difficulty stems from the fact that the three readers exhibit different concentration in the cell and manipulation of the cellular concentration of any of them (including deleting domains to elucidate function) causes disbalance in the entire 'reader' network and triggers some unspecific binding and overlap between the clients of each 'reader' (Shi et al, 2017). In the Discussion (p.10) we provide some discussion on the binding specificity and the link between the readers.

The YTH-based pulldown is helpful to identify unique mRNA clients of each 'reader', but does not reveal any mechanistic insights into the recognition mode. We believe that this is conferred not only by the 'reader' itself, but also specific signals embedded around the methylation sites (e.g. secondary structure and not sequence motifs) shape the specificity of the 'readers'. The specificity of binding will be a focus of intense research in the next years and most likely approaches based on global mRNA analysis would be helpful to elucidate the recognition motifs for each 'reader'.

Minor points:

1. Legend for Fig. 2C is missing.

We truly apologize for this! The legend should read:

"Average number of m⁶A modified DRACH motifs detected in the m⁶A-Seq of control cells or cells exposed to 500 μ M AS (stress). For comparison the average number of predicted DRACH motifs is included. $p = 1.49 \times 10^{-5}$ control vs stress, Mann-Whitney test."

Referee #3:

In this manuscript, Anders et al. claim that methylation of adenosine at N6 (m6A) represents a mechanism by which mRNAs are selectively routed into stress granules (SGs). Their claim is based on the observation that i) m6A-containing RNA can be visualized in SGs, ii) m6A increases in arsenite-treated cells, and iii) there is a partial overlap between SG-associated transcripts and m6A-modified transcripts. However, the evidence provided by the authors is by no means conclusive, and closer inspection of the

data suggests that there is actually no connection between m6A and SG-association. Importantly, there are no functional experiments that would directly show that m6A modification causes mRNAs to associate with SGs. To me, the conclusions drawn by the authors are fundamentally flawed, and not at all supported by their data.

Major concerns:

1) Localization of m6A in SGs: While the images in Fig. 1A/B clearly show that SGs contain m6A, this does not mean that m6A-modified mRNAs are enriched in SGs. In fact, the cytosolic m6A signal outside of SGs remains quite substantial. The authors would need to carefully quantify their IF analysis, and show that the m6A signal is enriched to a greater extent in SGs than the general poly(A) signal, which reflects the amount of mRNA in SGs. Such a thorough quantification of m6A intensities and comparison to a poly(A) signal has not been provided. At present, the m6A signal observed in SGs is a trivial result since most mRNAs are recruited to SGs, and m6A is known to be an abundant modification observed on a large proportion of mRNAs.

The Reviewer offers to quantify fluorescent images, which although done rarely in the published literature is greatly opposed in the scientific community, particularly when the fluorescent phenotype is mixed and comprises diffusive patterns and hyperfluorescent speckles. We understand the motivation of this Reviewer that quantification would provide easy means to assess differences among the images, but we are afraid that such comparison can only be performed when the fluorescent pattern is homogeneous and does not include hyperfluorescent species. Furthermore, we wish to emphasize that we use the fluorescent images only as observation which we back-up with other (sequencing) methods which allow quantitative comparisons.

Reflecting on the comment of the Reviewer, however, we believe that we should rephrase the text and provide better description of the fluorescent images. Namely, that the m⁶A antibodies do not discriminate between different types of methylation (m⁶A vs m⁷G cap is 2'-O-methyladenosine (A_m) which can be further methylated at N⁶ position (m⁶A_m)) and between single RNA entities and the signal is most likely dominated by rRNAs. Likely, the signal in the cytosol might be dominated from modified non-coding RNAs (tRNAs, rRNA of the hibernated 60S, for example). Using combined approach of PAR-CLIP to isolate SG clients together with m⁶A-Seq to identify their methylation pattern, we can group them by biotypes and quantitatively compare changes only on mRNAs.

2) Increase of m6A in arsenite-treated cells: By dot-blot (Fig. 1D), the authors show that there is a strong increase in m6A signal in arsenite-treated cells. My estimation by eye would be that this increase is 5- to 10-fold. However, this increase is in stark contrast to a) the lack of an increase of the m6A signal in the IF images (Fig. 1A), and b) the marginal increase in m6A/A of about 10% observed for mRNAs by m6A-Seq (Fig. 1E). In fact, the data suggest that the strong increase in m6A is due to methylation of an RNA species other than mRNA. I am concerned that different methods to detect m6A give such different results, and I am not convinced that there is an important change in methylation of mRNA in arsenite-treated cells.

The dot-blot reports on the methylation of total RNA which includes all RNA entities (mRNA, rRNA, tRNA, ncRNA...) which explains the higher signal intensity than those of the following figures which comprise the analyses of only one RNA biotype, mRNA. As we discuss in the manuscript on p. 5, the m⁶A antibodies do not discriminate between different types of methylation (m⁶A vs m⁷G cap is 2'-O-methyladenosine (A_m) which can be further methylated at N⁶ position (m⁶A_m)) and/or between single RNA entities. We believe that the m⁶A signal in the dot-blots is likely dominated by rRNAs which represent nearly 80% of all cellular RNAs. Thus, we performed m⁶A-seq, which allows for grouping the RNA entities by biotypes and for their separate analysis. In all our data analysis throughout the manuscript we analyze the methylation of mRNAs (including also Fig. 1E). Indeed, the methylation on mRNA is much lower than the methylation of non-coding RNAs which is also well represented in the literature.

The explanation in the text we offer to this is:

“The first nucleotide in the m⁷G cap is 2'-O-methyladenosine (A_m) which can be further methylated at N⁶ position (m⁶A_m) (Mauer et al, 2017), which can be also targeted by the m⁶A-antibodies. Furthermore, a large fraction of non-coding RNAs and rRNA from the small subunit within the initiation complex found in the SGs (Decker & Parker, 2012; Kedersha et al, 2013; Khong et al, 2017) can also be methylated and recognized by the m⁶A-antibodies. To discriminate internal m⁶A in mRNAs only, we performed global profiling of the RNA methylome in two independent biological replicates (m⁶A-Seq; (Meyer et al, 2012; Zhou et al, 2015)) under oxidative stress (500 μM) and compared the intrinsic m⁶A sites in mRNAs to those under permissive growth.” mRNA represents only a fraction from all detected m⁶A modified RNAs, but grouping the RNAs by biotypes allows to perform analysis only on a certain target group.

Furthermore, it should be noted that we weight or normalize the m⁶A sites equally (Fig 1E, 2B), as commonly established in the literature for transcription-wide analysis, to exclude any dominating effect of one transcript or even a methylation site of a transcript (see for example the single gene presentation (Fig 2D) which represent counts at different m⁶A with different methylation intensity). The changes we report are of the same order of magnitude what has been reported for changes in m⁶A pattern under other conditions (Zhou et al, Mol Cell 2018, Fig. 6A and 7A reporting on starvation stress; Zhang et al, Nature 2017; Fig. 1C representing changes upon abrogated methylation; Lin et al, 2017, Cell Res.; Fig. 4B; spermatogenesis in differentiation process; Coots et al, Mol Cell 2107; Fig. 3D reporting on METTL3 sensitivity).

Statistical analyses of all data support their significance.

3) The authors claim that "mRNAs sequestered in SGs exhibit a distinct m6A pattern". However, the m6A level of those transcripts that were found to be enriched in SGs (Fig. 2B) is virtually identical to the m6A level of total mRNA (Fig. 1E), and also very similar to the m6A level of those mRNAs that escape translational suppression (Fig. 3E). A conceptual problem with the author's hypothesis is that there is in fact rather little selectivity with regard to SGs: most if not all types of mRNAs were found to associate with SGs, yet the fraction of mRNA in SGs is different between genes. Hence, the analysis that the authors would need to do is a consecutive approach where they first isolate the SG-associated mRNA, and then measure the m6A level on this mRNA fraction in comparison to the m6A level on the fraction that remains cytosolic. The current analysis, which compares the list of genes whose mRNA contains m6A with the list of genes whose mRNA associates with SGs is not very meaningful.

Following harsh stress mRNAs can be found as translated in the cytosol or within the stress granules. The number of translated transcripts is low (108 in total). Hence, the signal of the total mRNAs is nearly identical to that of stress granules mRNAs which represent appr. 96% of all methylated mRNAs in the cell which provides an explanation for the similarity between Fig. 1E and 2B. The similarity between Fig. 1E and 2B are explained in the text (p. 6):

“The SG mRNA clients identified in the PAR-CLIP data set had significantly higher proportion of *de novo* methylation signals (Fig 2B) and these signals from these transcripts dominated the observed global increase of m⁶A signals in response to oxidative stress (Fig 1E).”

As explained below and also described in the caption to Fig. 3E, Mann-Whitney test reveals no significance of the difference (Fig. 3E) as opposed to the data represented in Fig. 1E and 2B.

The Reviewer argues that “most if not all types of mRNAs were found to associate with SGs, yet the fraction of mRNA in SGs is different between genes”, a comment which is based on a recent publication from the group of Roy Parker (2017 Moll Cell) where the mRNA clients of the stress granules were quantified using FISH-based assays combined with RNA-Seq. Unlike their approach, we do not quantify the copy number of each transcript, but use the sequencing to identify a transcript irrespective of the copy number it is present in the stress granules. The choice of this analysis is driven by the m⁶A-Seq approach which enables identification of methylation sites but does not bear information on

the quantity of methylation per transcript (all those technical features of the m⁶A-Seq including discussion on data processing are represented well in several reviews and methodological paper describing the m⁶A-Seq approach; please refer for example to the recent review by Groznic and Jaffrey, published in NCB 2018). This explains the comparison of gene lists.

4) mRNA is considered an essential component of SGs. The fact that knockdown of the METTL3 methyltransferase does not abolish SGs (as measured by G3BP1 staining, Fig. 1B) suggests that mRNAs are still recruited into SGs in the absence of METTL3. This directly contradicts the authors' hypothesis. The same is true for YTHDF3. Since its knockdown does not abolish SGs (Fig. 4A), YTHDF3 is most likely not important for recruitment of mRNAs into SGs.

We detect that 54% of all SG transcripts are additionally methylated under stress in the 5'UTR vicinity. The remaining 46% are unchanged, i.e. not additionally methylated under stress compared to the background. In the discussion part (p.10), we discuss possible mechanisms of selection of (additionally) methylated vs (not additionally) non-methylated under and propose scenarios which consider also the models from the literature (see the text below). Thus, we respectfully disagree with the Reviewer that our data contradict our hypothesis. Moreover, our data provide a comprehensive observation and explanation on the different modes on recruiting mRNAs into stress granules.

p. 10: "Our results demonstrate a prevalent stress-induced methylation in the 5'UTR and 5'vicinity of the CDSs, however not all cellular mRNAs we detected in the SGs are methylated (Fig 2E). Since oxidative stress induces stalling of ribosomes at initiation and/or early elongation (Fig. 2C), it is possible that ribosomes would sterically hinder the methylation in these regions. SGs are nucleated by translationally-stalled RNAs with assembled initiation factors which serve as scaffolds for RNA-binding proteins (Decker & Parker, 2012; Kedersha et al, 2013). Hence, the primary nucleation of SGs might occur in m⁶A-independent manner. Recent study shows that the core SG protein, G3BP1, is repelled by m⁶A (Edupuganti et al, 2017). It is conceivable that G3BP1 is recruited exclusively to mRNAs lacking m⁶A modification. Consistent with this model is our observation that silencing of the 'writer' complex alters only the association of m⁶A-modified mRNA and YTHDF3 with SGs, but not their size and number (Fig 1B and 4A)."

The reviewer states correctly that knockdown of YTHDF3 (or writers) does not abolish the stress granule formation. However it prevents the sequestration of the m⁶A methylated in stress granules unlike the non-methylated one (Fig. 4B). This data are in agreement with current models of stress granule formation as we discuss in the discussion (p. 10): "SGs are nucleated by translationally-stalled RNAs with assembled initiation factors which serve as scaffolds for RNA-binding proteins (Decker & Parker, 2012; Kedersha et al, 2013). Hence, the primary nucleation of SGs might occur in m⁶A-independent manner. Recent study shows that the core SG protein, G3BP1, is repelled by m⁶A (Edupuganti et al, 2017). It is conceivable that G3BP1 is recruited exclusively to mRNAs lacking m⁶A modification. Consistent with this model is our observation that silencing of the 'writer' complex alters only the association of m⁶A-modified mRNA and YTHDF3 with SGs, but not their size and number (Fig 1B and 4A)."

As mentioned by the Reviewer, YTHDF3 is found in the proteome of stress granules (this study and Jan et al, 2016 Mol Cell), but is not a scaffolding protein which are found in the core of the stress granules.

5) Lack of functional experiments: If the authors' hypothesis is correct, they should be able to show that the knockdown of METTL3 causes less mRNA to associate with SGs. This could be measured by poly(A) FISH as well as by SG purification followed by sequencing of SG-associated RNAs.

We thank the Reviewer for raising this point and in the revised manuscript we will include such experiment. Briefly, we analyze the partitioning of the m⁶A SG clients together with the general mRNA content of stress granules using polyA-FISH approach. We reason that silencing of *METTL3* would abrogate the colocalization of only methylated mRNAs (those 54.6 %) and would not influence the distribution of the remaining 46 % of mRNAs detected in stress-granules. Thus we isolate pull-down stress granules using G3BP1-specific antibodies and analyze the m⁶A signal with m⁶A-antibody and polyA signal using fluorescently labeled oligo-dT probes. Indeed, while the m⁶A signal is completely abrogated following *METTL3* silencing, the polyA-signal remains unchanged.

Fig. R2: Total RNA isolated from SG from control U2OS-G3BP1 cells or cells with siRNA knockdown of *METTL3*, *METTL3L14* and *WTAP* (-writers) and *YTHDF3* and detected with m⁶A antibody or oligo-dT pairing to polyA-tails of mRNAs.

Specific comments:

Fig. 1D: Does the increase in global m6A signal also occur with other types of stress? The authors show this only for arsenite treatment, yet in their title they indicate that m6A is a general mechanism by which mRNAs is triaged into SGs.

We observed an increase of m⁶A RNA modifications also under thermal stress (Fig. 1A). The focus on the manuscript is solely on oxidative stress and we only use the thermal stress example as an evidence that m⁶A signal colocalizes with stress granule independent of the stress type. Reflecting on the comment of the Reviewer, we offer to edit the text to clearly position the manuscript as focused on elucidating the mechanism by which mRNAs is triaged into SGs under oxidative stress.

Fig. 1E The number of m6A sites detected on mRNA by m6A-Seq increases only marginally by about 10% upon arsenite treatment, which is in stark contrast to the dramatic (5- to 10-fold) increase of the m6A signal on total RNA in Fig. 1D. Does this mean that other types of RNA (rRNA?) is subject to the dramatic increase in methylation?

See the explanation to main point 4. Indeed, the methylation of the rRNAs also change under stress, but this is outside the focus of the manuscript.

There is also a discrepancy between the m6A signal in the IF (Fig. 1A), which does not increase upon arsenite treatment, and the strong increase observed on total RNA (Fig. 1D). The measurements are in contradiction to each other.

As explained above, microscopy is not a quantitative approach (see p.1), particularly when hyperfluorescent spots are formed (e.g. stress granules due to a coalescence of fluorescent entities). Fig. 1D, as also explained also above (main point 4), represents the total RNA including RNAs residing in stress granules, cytosol and nucleus which precludes any direct comparison between Fig. 1A and 1D.

The Fig. 2B plot looks just like Fig. 1E, with a median of about 0.25 m6A/A sites. Hence, there is no enrichment of m6A sites on SG-associated transcripts, which is in direct contradiction to the authors's claim. How would this blot look for all the mRNAs that are not associated with SGs? Again just like Fig. 1E, I suppose.

The similarity between Fig. 1E and 2B are explained in the text (p. 6):

“The SG mRNA clients identified in the PAR-CLIP data set had significantly higher proportion of *de novo* methylation signals (Fig 2B) and these signals from these transcripts dominated the observed global increase of m⁶A signals in response to oxidative stress (Fig 1E).”

Or, when analyzing the m⁶A-Seq data we separate RNAs by biotypes and analyze only mRNAs. Following harsh stress mRNAs can be found as translated in the cytosol or within the stress granules. The number of translated transcripts is low (108 in total). Hence, the signal of the total mRNAs is nearly identical to that of stress granules mRNAs which represent appr. 96% of all methylated mRNAs in the cell.

Fig. 2E: The authors state that 57% of all SG-associated transcripts were methylated. What is the number for all the transcripts not associated with SGs? This comparison is crucial for the author's claim that methylation causes mRNAs to preferentially localize in SGs.

The number of all SG clients we identified is represented in Fig. 2E (6020 transcripts) and this is how the value of 54.7% (we believe the Reviewer means this with 57 %) is calculated and stated in the text on p.6 as following;
"Notably, 54,7% of all identified SG transcripts were methylated (Fig 2E)."

Fig. 3C: The authors state that "at mild stress, ...the majority [of transcripts] were stalled at initiation and early elongation" (page 7), while in Fig. 3C they show the ribosome footprint position of only two mRNAs. Where is the evidence that the majority of transcripts are stalled at initiation? Would it not be essential to compare the ribosome footprint position between control and stress conditions in a meta-gene analysis?

In response to this point, we will include metagene analysis in the revised version.

Fig. 3E: Why do the authors claim that the m⁶A signal of the translated mRNAs does not change upon arsenite treatment? The median goes up by about 10% just as for all mRNAs (Fig. 1E) or the SG-associated mRNAs (Fig. 2B). The fact that the difference is not significant in Fig. 3E is simply due to the fact the the number n is much smaller here. The comparison of these three plots is a good argument that there is really no connection between the m⁶A modification and triage of mRNAs between translation and stalling / SG-association.

We judge the data using statistical tests and in this case the Mann-Whitney test reveals no significance of the difference (Fig. 3E) as opposed to the data represented in Fig. 1E and 2B. For all analysis the p values are included in the corresponding figure legend. We kindly disagree with the Reviewer and object judging data by eye, and rely on statistical methods to justify differences.

EV2A: It is unclear what is depicted in this graph, in particular the first bar.

In this figure panel we show the mRNA levels of each gene using a log₂-fold change representation which is a common representation of qRT-PCR analysis or in general, of expression level analysis. In this analysis, as also explained in the figure legend, the level of each gene is normalized to a house-keeping gene, β-actin.

EV2A: In the Western blots, the kd samples need to be loaded on the same gel next to control samples. Otherwise, the signals cannot be compared.

In these experiments all samples were loaded on the same gel allowing their comparison. The vertical line designates that some lanes were omitted as they were either empty or loaded with samples that are irrelevant for the figure. If the manuscript is considered for publication, we will include those original images as usually requested by the journal.

EV3D: Are these really the motifs enriched in the SG-associated transcripts, or just the two most abundant motifs among the SG-associated transcripts? (This is not the same!) The first motif appears to be the canonical poly-A signal, hence it is trivial to find this as the most abundant motif among the SG-associated transcripts, because it is probably the most abundant motif in any (randomly chosen) subset of mRNAs.

The motifs represented in Fig. EV3D are the most common motifs in SG mRNA clients (see figure caption: "Two most abundant motifs among the SG mRNA clients revealed by

MEME motif search.”). We agree with the Reviewer that both motifs are typical RNA-binding motifs. Our reasoning for this analysis as explained in the text (p. 6) is to use it as a control to evaluate the specificity of the pull-down. Since we use anti-TIA antibodies, one may argue that we are extracting only TIA1-specific transcripts. TIA-1 binding motif is the polyA-binding motif. The nearly equal score of another C-rich motif suggests that through the 4sU crosslinking strategy we pull down not only TIA-clients. In the text we state:

p. 6: “Although we used anti-TIA1 antibodies to pull down SGs, the identified SG mRNA clients in the PAR-CLIP were enriched not only in TIA1-binding motifs (Fig EV3D), arguing that through the unspecific 4sU-mediated crosslinking we captured diverse SG mRNAs binding to different RNA-binding proteins.”

Reflecting on the comment of the Reviewer we believe that this analysis might be taken not only as a mere control of the pull-down but as de novo search for common motifs in stress granule clients. Thus, we offer to rephrase it and emphasize on the control character of this analysis.

Referee #1:

Comments for Anders et al.,

RNA modification at the N6 position of adenosines (m6A) is essential for viability of mice and plants. It is implicated in a variety of RNA processing events ranging from splicing, polyadenylation site usage to export, mRNA decay and translation. The methylation event is believed to be a co-transcriptional process as the methyltransferase METTL3 is nuclear. The modification is proposed to act as a dynamic mechanism by which the cell can tag transcripts emerging from the transcription machinery depending on cellular stress conditions like heat shock, UV irradiation or conditions that reduce transcription elongation. Furthermore, studies have shown that there is almost no difference in m6A methylation on transcripts associated to the chromatin and those found in the cytoplasm, under normal conditions (meaning the mark is added in the nucleus and not much of demethylation or methylation happens in the cytoplasm).

Anders et al addressed the role of m6A in moving mRNAs into Stress Granules (SGs) under stress induced by arsenite. They describe increased m6A methylation at the 5' end of transcript coding region and propose that it helps to triage mRNAs into SGs during stress. They implicate a role for an m6A reader protein YTHDF3 in selectively recognizing these 5'UTR marks.

Major concerns

1. Fig EV2B. Partitioning of METTL3 to the cytoplasm, but not of METTL14 etc. The suggestion that METTL3 alone is now going to be active in the cytoplasm is not possible based on structural and available biochemical data from the literature. Purified METTL3 alone is inactive. METTL14 is required to maintain the catalytic pocket by interacting with the "interface loop" of METTL3 (see the three structure papers). So the subsequent claim that under stress increased methylation is due to cytoplasmic methylation by mislocalized METTL3 is not supported. A simple way to rule this out is by blocking transcription. It is already known that increased m6A methylation can take place co-transcriptionally (Slobodin et al., 2017). The authors might also try labelling transcripts (SLAM-seq) at different time-points and follow their methylation profile.

We believe that the comment of the Reviewer might have been driven by our concise wording in this paragraph, since determining the cellular compartment in which methylation following stress exposure takes place is not central to this paper. With the colocalization experiment we aimed to address whether oxidative stress induces some gross changes in the expression of the readers and erasers, since disbalance may account for the differences in the methylation pattern. The alteration in the METTL3 partitioning we felt was worth reporting, as it resembled the METTL3 delocalization in lung cancer cells (Lin et al. 2016). However, we agree with the reviewer that METTL3 alone won't be able to catalyze methylation. We edited this part of the text (end of first paragraph, p. 6) and hope to have cleared up this misunderstanding.

The reviewer suggests some additional experiments to elucidate the precise timing and location of the m⁶A-modification following exposure to oxidative stress. As emphasized above, this is outside of the scope of this manuscript, but indeed such approaches may deliver interesting information and might be considered in the near future.

2. *Figure 2G: Molecular basis for targeting RNAs to SGs. This is the central question posed by the authors. The authors describe an increase in m⁶A at the 5'UTR and 5' vicinity of CDS. The experiment in Fig2G shows that a reporter carrying 102 nt of the CDS from ARL4C (a SG-enriched transcript) is enough to target the reporter to SG. This fusion of 102 nt was made to the coding region of the reporter, which now is extended at the N-terminus. Mutation of three DRACH motifs within this 102 nt stretch abolished localization. This shows that just three DRACH motifs in the N-terminus of the coding sequence is sufficient for targeting. How general is this observation? How many cells show this co-localization pattern? Are these three sites methylated in the context of the reporter? Will presence of just three additional motifs (not the entire 102 nt) in fusion to the reporter be sufficient? I am trying to understand the nature of the signal for triaging to the SG? What about m⁶A on the 5'UTR? This experiment may be exploited further to understand the molecular basis for this observation.*

From the m⁶A-Seq data set we selected transcripts which were additionally modified in the 5' vicinity of their CDSs. Our rationale was to use a natural sequence upstream of a reporter construct to specifically target the role of those stress-induced modifications. In the particular case of ARL4C, the prevalent stress-induced m⁶A takes place within the first 102 nt, hence our choice for this exact fragment. The question of how many m⁶A-sites are necessary for triaging into SGs is in part irrelevant here, as we rarely see a transcript that has a single modification site. Furthermore, in our data set not every transcript methylated in the 5' termini of the CDSs was methylated in the 5'UTR. The methylation of the 5' termini of the CDS was more pervasive than combined with the 5'UTRs. In the figure legend, we have added the information on the number of cells (83%) from the whole batch we screened which show that lack of m⁶A abrogates the localization of this mRNA in SGs.

How to explain SG localization of RNAs that are not methylated? Or SG localization of those RNAs which are methylated?

As we extensively explain in the manuscript (see the discussion and Fig. 5), for a larger fraction of mRNAs (appr 55 %) stress-induced m⁶A modifications in the 5' vicinity of the transcripts serve as a specific mechanism for triaging them into SGs. Another fraction of mRNAs (appr. 45%), which are not methylated, most likely associate with the SGs triggered by the oxidative stress-induced stalling at initiation which is in line with previous publications. The proposed model in Fig. 5 adequately represents our data and summarizes both options for deposition of mRNAs into SGs.

3. *Figure 4: Role of YTHDF3 in triaging mRNAs into SG. It is hard for this reviewer to comprehend how it can carry out this function of recognizing m⁶A on the 5' CDS regions. How is it different from the other YTH proteins? There is accumulated evidence from a number of studies that the three YTHDF proteins (YTHDF1-3) are in fact largely redundant, and share the same binding sites across the transcriptome (Patil et al., 2018).*

Our knowledge on the recently discovered family of the 'reader' YTH-domain proteins is constantly growing: published literature along with our study clearly show a functional 'specialization' of the three readers, with YTHDF1 specifically binding to the 5'UTRs (Li et al, 2017; Shi et al, 2017), YTHDF2 specifically targeting the 3'UTRs and triaging mRNAs for degradation (Meyer et al. 2012; Wang et al, 2014), and YTHDF3 being active under stress response (this study). While the functional specialization of the three readers is undisputable, the molecular mechanisms on this specialization remain elusive and will be likely intensively addressed in the next few years. Indeed, the Reviewer is correct about the functional redundancy among the three YTHDF-readers. This conclusion stems from the fact, that manipulation of the cellular concentration of any of them (including deleting domains to elucidate function) causes disbalance in the entire 'readers' network and triggers some unspecific binding and overlap of their clients or binding sites (Shi et al, 2017). In the Discussion (p.12/p.13) we provide some discussion on the binding specificity and the interplay between the readers.

Minor

1. *Figure legends 2G: "Deletion of the methylation sites in the 5' vicinity of ARL4C mRNA renders its localization into SGs. Renders should be hindens?"*

Corrected.

2. *Somewhere else in the text "Motives" should be motifs.*

The misspelling was corrected.

3. *Figure 2G: The reporter assay. The text in page 7 mentions YFP fusion, while legends and methods mentions CFP. Correct it.*

We apologize for the mislabeling of CFP and have corrected this.

Referee #2:

The authors have performed additional experiments to strengthen their findings and conclusions. Their new information and revised manuscript have answered most of my previous questions with regard to stress-inducible m6A. While no mechanistic insight for YTHDF3 is provided, I think the concept of m6A facilitated mRNA triaging is better demonstrated in this version of manuscript. Thus, I recommend the publication of this paper.
We thank the Referee for the positive assessment.

2nd Editorial Decision

25 June 2018

Thank you for submitting your revised manuscript entitled "Dynamic m6A methylation facilitates mRNA triaging to stress granules". I appreciate the introduced changes, and I am happy to accept your manuscript in principle for publication in Life Science Alliance.
